



# IPSL-CM5A2. An Earth System Model designed for multi-millennial climate simulations

Pierre Sepulchre[1], Arnaud Caubel[1], Jean-Baptiste Ladant[1,2], Laurent Bopp[3], Olivier Boucher[4], Pascale Braconnot[1], Patrick Brockmann[1], Anne Cozic[1], Yannick Donnadieu[5], Victor Estella-Perez[6], Christian Ethé[8], Frédéric Fluteau[7], Marie-Alice Foujols[8], Guillaume Gastineau[6], Josefine Ghattas[8], Didier Hauglustaine[1], Frédéric Hourdin[4], Masa Kageyama[1], Myriam Khodri[6], Olivier Marti[1], Yann Meurdesoif[1], Juliette Mignot[6], Anta-Clarisse Sarr[5], Jérôme Servonnat[1], Didier Swingedouw[9], Sophie Szopa[1], and Delphine Tardif[7]

[1]Laboratoire des Sciences du Climat et de l'Environnement, LSCE/IPSL, CEA-CNRS-UVSQ, Université Paris-Saclay, F-91191 Gif-sur-Yvette, France
[2]Department of Earth and Environmental Sciences, University of Michigan, Ann Arbor MI 48109, USA
[3]Laboratoire de Météorologie Dynamique, Sorbonne Université / CNRS / École Normale Supérieure, Paris, France
[4]Laboratoire de Météorologie Dynamique, IPSL, CNRS, Sorbonne Université, Paris, France
[5]CEREGE, Aix-Marseille University, CNRS, IRD, Coll. France, INRA, Technopole Arbois-Méditerranée, BP80, 13545 Aix en Provence cedex 4, France
[6]Laboratoire d'Océanographie et du Climat : Expérimentations et Approches Numériques, Sorbonne Université / CNRS, 4 place Jussieu, 75252 Paris Cedex 05, France
[7]Université de Paris, Institut de physique du globe de Paris, CNRS, F-75005 Paris, France
[8]Institut Pierre-Simon-Laplace, Sorbonne Université/CNRS, Paris, France
[9]Environnements et Paléoenvironnements Océaniques et Continentaux (EPOC), UMR CNRS 5805, EPOC, OASU, Université de Bordeaux, Allée Geoffroy Saint-Hilaire, Pessac 33615, France

**Correspondence:** Pierre Sepulchre (pierre.sepulchre@lsce.ipsl.fr)

**Abstract.** Based on the CMIP5-generation previous IPSL earth system model, we designed a new version, IPSL-CM5A2, aiming at running multi-millennial simulations typical of deep-time paleoclimates studies. Three priorities were followed during the set-up of the model: (1) improving the overall model computing performance, (2) overcoming a persistent cold bias depicted in the previous model generation, and (3) making the model able to handle the specific continental configurations of the geological past. Technical developments have been performed on separate components and on the coupling system to speed up the whole coupled model. These developments include the integration of hybrid parallelization MPI-OpenMP in LMDz atmospheric component, the use of a new library to perform parallel asynchronous input/output by using computing cores as "IO servers", the use of a parallel coupling library between the ocean and the atmospheric components. The model can now simulate ∼100 years per day, opening new possibilities towards the production of multi-millennial simulations with a full earth system model. The tuning strategy employed to overcome a persistent cold bias is detailed. The confrontation of an historical simulation to climatological observations shows overall improved ocean meridional overturning circulation, marine productivity and latitudinal position of zonal wind patterns. We also present the numerous steps required to run the IPSL-CM5A2 for deep-time paleoclimates through a preliminary case-study for the Cretaceous. Namely, a specific work on the ocean model grid was required to run the model for specific continental configurations in which continents are relocated



according to past paleogeographic reconstructions. By briefly discussing the spin-up of such a simulation, we elaborate on the requirements and challenges awaiting paleoclimate modelling in the next years, namely finding the best trade-off between the level of description of the processes and the computing cost on supercomputers.

## 1  Introduction

Despite the rise of high-performance computing (HPC), the ever-growing complexity and resolution of General Circulation
Models (and subsequent Earth System Models –ESMs-) have restricted their use to rather short "snapshot" integrations of the paleoclimates, with durations ranging from decades to a few thousands of years. Longer simulations of several thousands of years are yet often desirable, in particular in paleoclimate studies, in order to either (i) reach a fully equilibrated deep ocean when initialized from idealized thermohaline conditions (a typical procedure in deep-time, pre-Quaternary, paleoclimate simulations) or (ii) address multi-millennial transient climate evolution such as glacial-interglacial cycles of the Quaternary
period, but also to (iii) project millennial-scale future climatic trends. Through years, several strategies have been set up to overcome this issue:

- Earth System Models of Intermediate Complexity (Claussen et al., 2002), which coarse spatial resolution and simplifications in the physics allow very efficient computation times, have been used extensively to explore Quaternary paleoclimates and run transient simulations (Caley et al., 2014). As an example, the LOVECLIM model (Goosse et al.,
2010), that provides more than 1000 simulated years per day (hereafter SYPD) with a ∼5.6 ° resolution and 3 vertical levels in the atmosphere (Bouttes et al., 2015b) has been used repeatedly to study the last interglacial (Goelzer et al., 2016).

- Regarding GCMs, early methods to circumvent the constraint of computation time have involved (i) using atmospheric-only models forced by prescribed sea-surface temperatures (e.g. (Roberts et al., 2014)), (ii) coupling the atmospheric
component to mixed-layer ocean models with prescribed heat transport –i.e. slab ocean models- (Otto-Bliesner and Upchurch, 1997), and (iii) asynchronous coupling between atmospheric and ocean models (Bice et al., 2000). If efficient in terms of computation time, such strategies prevented any assessment of ocean dynamics and associated feedbacks on the climate system. Later, experimental designs with GCMs for deep-time paleoclimate modelling relied on simulations "branched" on each other to make the ocean reach different equilibria in a reasonable amount of time (namely less than
1000 years, (Liu et al., 2009)).

- Versions of fully-coupled GCMs have been maintained at low spatial resolution, allowing long integrations in a reasonable amount of wallclock time. Examples include the FAMOUS (FAst Met Office/UK Universities Simulator) model, a derivative of HadCM3 GCM that runs at a 7.5° longitude x 5° latitude resolution with 11 levels in the atmosphere and averages 100 SYPD, and that is frequently used for paleoclimate (e.g. (Gregoire et al., 2012; Roberts et al., 2014) and
future climate studies (Bouttes et al., 2015a). The FOAM (Fast Ocean Atmosphere Model) model is another example of this class of GCM. It couples the NCAR CCM2 (7.5° x 4.5° and 18 vertical levels but an updated physics equivalent to



the NCAR CCM3 model) atmosphere model and an ocean model similar to GFDL MOM. It reaches an average of 250 SYPD and has been routinely used for almost twenty years in numerous deep-time paleoclimate studies (Brown et al., 2009; Donnadieu et al., 2006; Ladant et al., 2014; Pohl et al., 2014; Poulsen et al., 2001; Roberts et al., 2014). NCAR has also developed a low-resolution (T31 –i.e. approximately 3.75° longitude x 3.75° latitude, 26 vertical levels, 35 SYPD) version of CCSM3 GCM (Yeager et al., 2006) that provided "the first synchronously coupled atmosphere-ocean general circulation model simulation from the Last Glacial Maximum to the Bølling-Allerød (BA) warming" (Roberts et al., 2014) as well as transient Holocene simulations (Liu et al., 2014). More recently, this group has released a similar version for CCSM4 (Shields et al., 2012)), at the same "low resolution", with performance reaching 64 SYPD. This model has been used for multi-millennial fully-coupled paleoclimate simulations (Brierley and Fedorov, 2016; Burls et al., 2017). A similar strategy has been applied with the ECHAM5/MPI-OM model (formerly known as COSMOS), which has been used at T31 resolution in the atmosphere for long paleoclimate simulations of the Holocene (Dallmeyer et al., 2017; Fischer and Jungclaus, 2011), and has been recently implemented with water isotopes (Werner et al., 2016).

– For transient experiments, i.e. with an evolving insolation, Lorenz and Lohmann (2004) proposed an accelerated insolation forcing. When the simulation calendar advances by one year, the calendar for the computation of the incoming insolation advances by 10 to 100 years. The method assumes that the fast components of the climate system (atmosphere, upper ocean) are in constant equilibrium with insolation, which has much longer timescale. It is suitable only for periods of time with almost no change for the slow components (deep ocean, land ice). This method is easily implementable in any climate model (e.g., Crosta et al., 2018). Recently, "continuous" transient simulations have been run through the use of a statistical emulator, calibrated using several steady-state GCM simulations, with varying orbital configurations and atmospheric pCO2 concentrations (Lord et al., 2017).

The original IPSL-CM5A-LR (we will omit the -LR suffix in the following) model was developed and released in 2013 "to study the long-term response of the climate system to natural and anthropogenic forcings as part of the 5th Phase of the Coupled Model Intercomparison Project (CMIP5)" (Dufresne et al., 2013). This model, with an atmospheric resolution of 3.75 x 1.875° in longitude-latitude and 39 vertical levels, and of 2 degrees and 31 vertical levels in the ocean, has been used for several paleoclimate studies (e.g., Kageyama et al., 2013; Roberts et al., 2014; Tan et al., 2017; Zhuang and Giardino, 2012), including studies benefiting from the explicit representation of marine biogeochemistry (Bopp et al., 2017; Ladant et al., 2018; Le Mézo et al., 2017), vegetation dynamics and land biosphere (Tan et al., 2017). Still, IPSL-CM5A computation time, which averages 10 SYPD, has hindered its use for multi-millennial experiments that are typical for Quaternary or "deep-time" paleoclimate studies in which a fully-equilibrated deep ocean is mandatory. In this paper, we present and evaluate IPSL-CM5A2, a CMIP5-class ESM, including interactive vegetation and carbon cycle, and designed for multi-thousand years experiments.





## 2 From IPSL-CM5A to IPSL-CM5A2

Apart from better computing performances, setting up IPSL-CM5A2 aimed at reducing two of the main biases of IPSL-CM5A. First, an important cold bias has been depicted in global surface air temperature (t2m) in IPSL-CM5A (Dufresne et al., 2013).

This cold bias is associated with a position of the atmospheric eddy driven jets too much equatorward beyond the spread of other CMIP5 models, especially in the Northern Pacific and Southern Hemisphere (Barnes and Polvani, 2013). Moreover, the Atlantic Meridional Overturning Circulation (AMOC) has maximum values between 8 and 10 Sv in historical simulations with IPSL-CM5A, a value lower than the observation range and below the values obtained in the other CMIP5 models (Zhang and Wang, 2013).

IPSL-CM5A2 development was initiated in 2017 on the supercomputer CURIE, operated at the Très Grand Centre de Calcul (TGCC), Bruyères-le-Châtel, France, as a part of the GENCI French computing consortium. CURIE offered different fractions of x86-64 computing resources, including 5040 "thin" nodes (BullX B510). Each node was made of two 8-core 2.7 GHz Intel Sandy Bridge processors, providing a total of 80 640 cores for computation. CURIE was replaced by the supercomputer JOLIOT-CURIE in July 2018. On this latter machine, IPSL models are run on a partition of 1 656 nodes, each node made

of two 24-core 2.7 GHz Intel Skylake processors, providing a total of 79 488 cores for computation. Here we depict how IPSL-CM5A2 differs from IPSL-CM5A in terms of components (2.1) and technical characteristics (2.2). Then we present the computing performances of the new model, first on CURIE compared to IPSL-CM5A as it was originally conducted, then on JOLIOT-CURIE, the TGCC machine that will be used in the next years by the IPSL climate model community and on which we conducted a set of scaling experiments (2.3).

## 2.1 IPSL-CM5A2 components: evolution from IPSL-CM5A

IPSL-CM5A2 is built on updated versions of IPSL-CM5A components, and is configured at the same spatial resolutions, i.e. the regular 3.75 x 1.875° longitude-latitude grid and 39 vertical levels for LMDZ, and the ORCA2 curvilinear grid for NEMO, which is 2 degrees with a refinement to 0.5° in the tropics, and 31 vertical levels (Fig.1). This choice results from an early evaluation of the performances presented in SI1. Here we target the significant differences between the two models

and provide a brief summary of each component. We refer the reader to Dufresne et al. 2013 for more detailed description of LMDZ, NEMO and ORCHIDEE characteristics.

### 2.1.1 LMDZ

LMDZ is the atmospheric general circulation model developed at Laboratoire de Météorologie Dynamique, France. LMD models have been the atmospheric component of the IPSL coupled models since the earliest IPSL coupled model, that in-

cluded LMD5.3 (Braconnot et al., 1997). LMDZ is based on the coupling of a dynamical core in which primitive equations of meteorology are discretized, and a set of physical parameterizations. Here, we use the version LMDZ5A, where A designates the so-called "standard physics" (SP) for the physical atmospheric parameterizations, also used in IPSL-CM5A. The choice of keeping this CMIP5-like physical settings and not benefiting from improvements in the representation of convective boundary

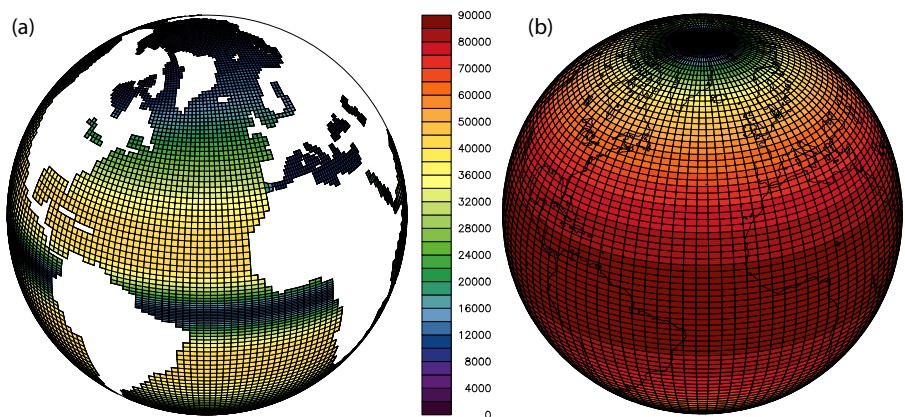

**Figure 1.** NEMO (a) and LMDZ (b) grid cell areas, in km$^2$, showing the refinements of the ocean grid in the tropics and over the Mediterranean Sea as well as the refinement of the atmospheric grid to the pole.

layer, cumulus clouds and the diurnal cycle of continental convective rainfall included in the "New Physics" (NP) is driven
both by (i) the fact that no coupled version of IPSL including the new parameterizations was available when IPSL-CM5A2
project was initialized and (ii) computing cost. The NP parameterizations in LMDZ have been validated for higher vertical
resolution (79 levels) and require higher-frequency time coupling between the physics and the dynamics. SP indeed requires
a time step for the dynamics of 3 minutes and a coupling every 30 minutes with the physics, whereas dynamics is computed
every 2.15 minute and coupled every 15 minutes in NP, leading to significant increase in computation time (see next section). In
terms of revision nomenclature, IPSL-CM5A included LMDZ5, revision 2063, whereas IPSL-CM5A2 includes revision 3342.
This represents a >2-year gap between both versions.The differences mostly include bug corrections, various improvements
in the physics/dynamics interface, and energy conservation. Changes also concern the Emanuel convection scheme, with bug
corrections and changes in the upper bound of the deep convection loop.

### 2.1.2 ORCHIDEE

ORCHIDEE (Krinner et al., 2005) is a dynamic global vegetation model (DGVM) that can run either in stand-alone mode,
coupled to LMDZ, or included as the land surface component in the IPSL climate model. ORCHIDEE is made of three parts : (i)
the "hydrology module" SECHIBA (Schématisation des Echanges Hydriques à l'Interface entre la Biosphère et l'Atmosphère)
(De Rosnay and Polcher, 1998; Ducoudré et al., 1993), (ii) parameterizations regarding vegetation dynamics based on plant
functional types (PFTs) and derived from the LPJ DGVM (Sitch et al., 2003), and (iii) a "carbon module" called STOMATE
(Saclay Toulouse Orsay Model for the Analysis of Terrestrial Ecosystems), that simulates vegetation phenology and carbon
dynamics. SECHIBA "describes exchanges of energy and water between the atmosphere and the biosphere, and the soil water
budget" (Krinner et al., 2005) with a time step of 30 minutes. In IPSL-CM5A and IPSL-CM5A2, SECHIBA includes a two-
layer soil hydrology scheme, in which maximum water storage capacity is set globally to 300 kg m-2 over a two-meter soil





depth (Guimberteau et al., 2014). Water is redistributed between the two layers through a downward flux parameterized the
early ideas of Choisnel (Choisnel et al., 1995; Ducharne et al., 1998). Rain falling from the canopy feeds the upper layer
that loses water both by root extraction and soil evaporation, whereas water storage in the bottom layer decreases only as
a function root extraction (Guimberteau et al., 2014). When total soil moisture storage reaches the maximum water storage,
the excess water amount is converted to runoff. STOMATE links the fast hydrologic and biophysical processes to the slow
processes of vegetation dynamics from LPJ. It includes formulations for photosynthesis, carbon allocation, leaf phenology, and
respiration (autotrophic and heterotrophic). Vegetation is represented through 13 plant functional types (PFTs), one including
bare soil. Each PFT is characterized by specific parameters controlling their dynamics, i.e. climatic control of vegetation
establishment, elimination, light competition, fire occurrence and tree mortality. IPSL-CM5A2 includes the version 3930 of
the ORCHIDEE model, whereas IPSL-CM5A was based on an older tagged version (ORCHIDEE 1_9_5, for the record).
From the technical point of view, these two versions differ in that the IPSL-CM5A2 version of ORCHIDEE benefits from
XIOS for output diagnostics and from hybrid (MPI-OpenMP) parallelization, as for LMDZ. Amongst several bug corrections,
continental evaporation computation has been corrected to ensure a full closure of the water budget.

### 2.1.3 NEMO-LIM2-PISCES

NEMO has been upgraded from v3.2 to v3.6. River runoffs are now added through a non-zero depth, and have a specific
temperature and salinity. Although a new version of the sea-ice model (LIM3, (Rousset et al., 2015)) has been developed and
is included in the 6th version of IPSL model (IPSL-CM6A, Servonnat et al. in prep), the coupling with the ocean model on the
ORCA2 mesh was not ready and we chose to keep the LIM2 configuration for IPSL-CM5A2. The biogeochemical component
of NEMO is PISCES (Pelagic Interaction Scheme for Carbon and Ecosystem Studies) and has been upgraded from PISCES-v1
(Aumont and Bopp, 2006) in IPSL-CM5A to PISCES-v2 (Aumont et al., 2015) in IPSL-CM5A2. PISCES simulates the ocean
carbon cycle and includes a simple representation of the lower trophic level ecosystem, with four plankton functional types (2
phyto and 2 zoo) and the cycles of the main oceanic nutrients (N,P,Si,Fe). Without any change for the general model structure
(with 24 state variables), the transition from PISCES-v1 to PISCES-v2 does include a number of developments, e.g. on the
iron cycle, on zooplankton grazing, on dissolved and particulate organic matter cycling (see Aumont et al., 2015 for a detailed
description).

## 2.2 Technical developments

Technical developments have been performed on both individual components and the coupling system to accelerate the entire
coupled model.

### 2.2.1 OASIS3-MCT coupling library and interpolation scheme for ocean-atmosphere exchanges

The ocean-sea ice-atmosphere coupling is close to the one used in IPSL-CM5A (Dufresne et al., 2013), except the coupling
system has been switched from OASIS3.3 to OASIS3-MCT (for Model Coupling Toolkit, Valcke, 2013), that constitutes a





parallel coupling library used in both NEMO and LMDz components. It ensures fully parallel interpolation and exchange of
the coupling fields. As in IPSL-CM5A, 24 coupling fields are exchanged, 7 from the ocean to the atmosphere and 17 from the
atmosphere to the ocean (Valcke, 2013).

### 2.2.2   XIOS library

The use of XIOS as input-output library allows performing parallel asynchronous input/output by using computing cores as
"IO servers". These servers are dedicated to the reading and the writing of the data and permit model computation to continue
while data is written or read.

### 2.2.3   Integration of hybrid parallelization MPI-OpenMP in LMDz atmospheric component

In LMDz, longitudinal filtering is applied on the dynamical equations for latitude higher than 60° in both hemispheres (Hourdin
et al., 2013). Thus the choice of MPI domain decomposition is optimal only along latitudinal bands. This decomposition has
been initiated in the LMDz version of IPSL-CM5A with a minimum of three latitude bands per MPI process, reduced to two
in IPSL-CM5A2. In addition, the use of multi-core processors in HPC led us to add a shared memory parallel processing
(OpenMP) in LMDz. The use of this shared memory parallelism on vertical levels in the dynamics allows to increase the
number of cores used on the supercomputer and consequently to reduce the elapsed time of the simulation (see next section).

## 2.3   Computing performances

### 2.3.1   IPSL-CM5A2 vs IPSL-CM5A: computing performances on supercomputer CURIE

This section assesses how the technical developments depicted in section 2.2 improve the computing performances of the
model. From (Balaji et al., 2017) we use a set of metrics relevant for studying computational performances of IPSL-CM5A2.
Besides SYPD, four metrics are used here:

–   CHSY (core-hour per simulated year): This is the product of the model elapsed time for 1 simulated year times the
180        number of computing cores. It indicates the computing cost of a simulation.

–   Parallelization: number of computing cores allocated for the simulation.

–   Coupling cost: overhead due to the coupling, computed as the cost of the coupling algorithm itself, i.e. the ratio of CHSY
with and without coupling (CHSY – CHSY no coupling) / CHSY. The "no coupling" means atmospheric model only.

–   Data output cost: cost of performing I/O, computed as the ratio of CHSY with and without I/O : (CHSY – CHSY no IO)
185        / CHSY.

Compared to IPSL-CM5A, which could be run on 38 cores only, hybrid parallelization for IPSL-CM5A2 permitted to design
two configurations using more than 100 cores: a high throughput version (IPSL-CM5A2-T) on 160 cores, and a "fast" version



| | Configuration | SYPD | CHSY | Parallelization | Coupling | I/O |
|---|---|---|---|---|---|---|
| | | | | (TOTAL (ATM+OCE+CPL)) | | |
| CURIE | IPSL-CM5A | 10 | 91 | 38 (32+5+1) | 12% | 10% |
| | IPSL-CM5A2 (T) | 38 | 101 | 160 (32x4 + 31 + 1) | <1% | 10% |
| | IPSL-CM5A2 (S) | 56 | 129 | 302 (32x8 +45 +1) | <1% | 10% |
| JOLIOT-CURIE | IPSL-CM5A2 | **80** | **91** | **302 (32x8 +45 +1)** | <1% | - |
| | | 81 | 94 | 317 (32x8 +60 +1) | <1% | - |
| | | **98** | **107** | 437 (47x8 +60 +1) | <1% | - |
| | | 98 | 110 | 452 (47x8 +75 +1) | <1% | - |
| | | 103 | 145 | 625 (47x12 +60 +1) | <1% | - |
| | | 104 | 148 | 640 (47x12 +75 +1) | <1% | - |

**Table 1.** Computing performances metrics collected on CURIE and JOLIOT-CURIE platforms. Parallelization is defined as the total number of computing cores (Number of MPI (x Number of OMP) per component). One core is always used for XIOS process.

(IPSL-CM5A2-S) running on 302 cores. IPSL-CM5A2-T configuration allows to reach 38 SYPD (3.8 times faster than IPSL-CM5A) with a 10.8 % increase in computing cost (CHSY, Table 1 and Fig. 2). IPSL-CM5A2-S reaches 56 SYPD, for a CHSY
42 % larger than for IPSL-CM5A (129,000 hours for 1,000 simulated years). Besides, CHSY is greater for IPSL-CM5A2-S, which means there is a better use of a resource allocation than with IPSL-CM5A. The replacement of the sequential OASIS3.3 coupler by the parallel OASIS3-MCT library in IPSL-CM5A2 also allows reducing the cost of the coupling from 12 to 1 % of the total CPU time of a simulation. Regarding inputs/outputs aspects, IPSL-CM5A used the IOIPSL library, which performed sequential writing and imposed a rebuild post-processing step to merge multiple files into one single file. Instead, IPSL-CM5A2
uses the XIOS library that writes out data asynchronously using dedicated I/O servers. Parallel writing allows users to obtain directly one single file without any time-consuming rebuild step. Still, although the writing is performed asynchronously, the I/O cost remains important in IPSL-CM5A2 (10 %) due to operations (temporal operations, variables combinations) that are now performed by the library (on the client side, i.e. model process) rather than previously into the model.

### 2.3.2 Performances of IPSL-CM5A2 on the supercomputer JOLIOT-CURIE

Switching from CURIE to JOLIOT-CURIE supercomputer opened opportunities to increase performances of IPSL-CM5A2 through several aspects: first, replacing Intel Sandy Bridge by more recent Intel Skylake processors was expected to reduce computation time. Second, we decided to test higher parallelization levels, by combining (i) two-latitude band parallelization (instead of three), (ii) an increased number of OMP processes for LMDz, and (iii) more MPI processes for NEMO. Figure 2 depicts the scaling of IPSL-CM5A2 on JOLIOT-CURIE and CURIE. It shows that for the same parallelization set at 302
cores, IPSL-CM5A2 computes about 43% faster on JOLIOT-CURIE than on CURIE (80 SYPD compared to 56 SYPD), with a cost reduced by ∼30 % (91,000 hours for 1000 year). Finding the most valuable configuration requires both to find a tradeoff between maximizing SYPD and minimizing CHSY for each component and to minimize each component waiting time in coupled configuration. To achieve the latter goal, we used LUCIA (Maisonnave and Caubel, 2014) to determine LMDZ and



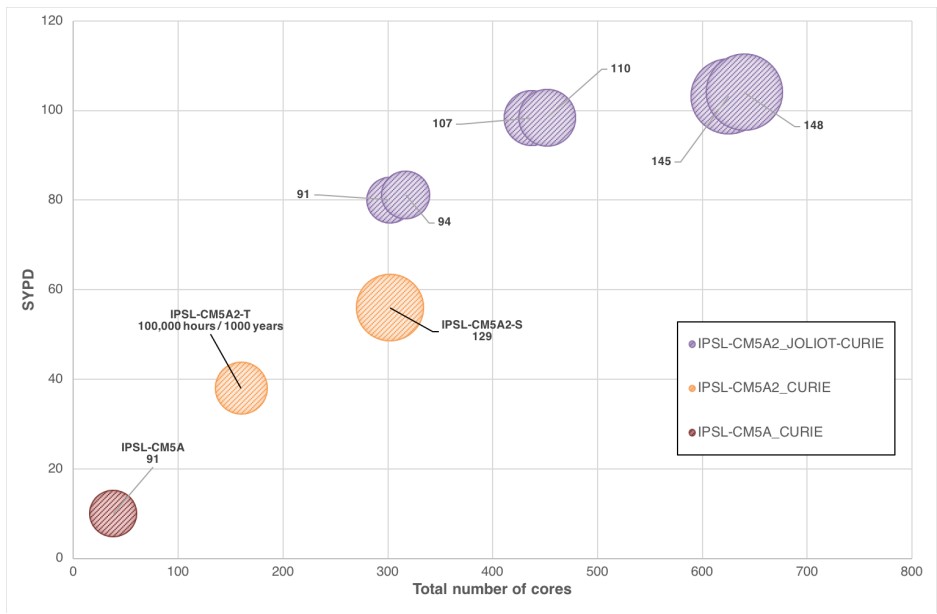

**Figure 2.** Scalability, i.e. SYPD as a function of parallelization, of IPSL-CM5A2 on CURIE and JOLIOT-CURIE supercomputers. The surface of the disks is proportional to the computing cost (CHSY).

NEMO computing and waiting time. From the 302-cores configuration, we made a step-by-step increase in LMDZ and NEMO
resources that shows that both components waiting times are minimized when LMDZ runs on 47 MPI processes and 8 OMP
threads together with NEMO using 60 MPI processes (Fig. A2). This configuration uses 437 cores and allows simulating
98 years per day, leading to a cost of 107,000 hours for a thousand-year simulation. Scaling up does not provide significant
improvements in SYPD numbers but the cost of the simulation is dramatically enhanced (39 % increase when we dedicate more
than 600 cores to the model (Fig. 2 and Table 1). Consequently, the 437-core configuration has been chosen as the distributed
version of reference.

## 3   Tuning IPSL-CM5A2

IPSL-CM5A2 is evaluated through two sets of simulations. First, we describe the setup of a steady-state pre-industrial simu-
lation that involved several experiments with tuning steps and bug fixes. This setup is used to discuss the tuning strategy, the
model large-scale characteristics, ocean spinup, energy and freshwater budgets, as well as multi-centennial ocean variability.
Second, a transient "historical" simulation, forced by boundary condition between years 1850 and 2005, has been designed to
evaluate the model with respect to observations (section 4).





### 3.1 Tuning: target and strategy

We focus here on the general tuning strategy adopted by defining the target and how we reached it, rather than giving a comprehensive description of the numerous simulations that have been designed to obtain the final pre-industrial simulation.

As depicted in Hourdin et al. (2017), several choices need to be made when tuning a model, namely defining a relevant target and associated metrics, defining in what mode the model is to be tuned (e.g. atmospheric only, fully coupled, etc.), and choosing which parameters will be used to reach the target. As most of modeling groups (Hourdin et al., 2017) our choice was to tune IPSL-CM5A2 through pre-industrial runs, using alternatively atmospheric and fully-coupled experiments. A first untuned 1,000-year spin-up of IPSL-CM5A2 forced by CMIP5 pre-industrial boundary conditions provided an adjusted global

surface air temperature of 11.3°C. Such a cold bias had been depicted in the previous IPSL-CM5A (Dufresne et al., 2013). We targeted to increase global-mean surface temperature by 2.2°C to reach 13.5°C in pre-industrial conditions with IPSL-CM5A2, expecting this value to translate into 15.5°C in simulations with present-day conditions.

Regarding the free parameters, the choice was to use a parameter which controls the conversion of cloud water to rainfall to alter the cloud radiative forcing (CRF), and thereby the TOA net radiation ($Q_{TOA}$, counted positive downward) and the global-

mean surface temperature. Using clouds parameters for tuning, the most uncertain parameters that affect the most radiation, is also well shared practice among modeling groups. Previous results obtained with LMDZ and IPSL-CM5A show that changing TOA balance by 1 W.m$^{-2}$ results in a change in temperature of 1 K. It is also the typical value of the sensitivity of global temperature to greenhouse gas concentration. Our +2.2°C target thus translates into an increase of $Q_{TOA}$ by 2.2 W.m$^{-2}$. In LMDZ, the conversion of cloud water to rainfall is computed following Sundqvist (1978), as

$$\frac{dq_{lw}}{dt} = -\frac{q_{lw}}{\tau_{conv}} \left[ 1 - e^{-(\frac{q_{lw}}{c_{lw}})^2} \right] \tag{1}$$

where $q_{lw}$ is the mixing ratio,

$c_{lw}$ is the in-cloud water threshold for autoconversion, set at 0.418 g.kg-1 in IPSL-CM5A,

For $q_{lw} \gg c_{lw}$ the time constant for auto-conversion is $\tau_{conv}$ (here set at 1800 s) while it goes to infinity (no auto-conversion)

for $q_{lw} \ll c_{lw}$.

Decreasing $c_{lw}$ is expected to lower cloud water content and reduce the CRF. First, we carried out atmospheric-only simulations with varying $c_{lw}$ values to define the sensitivity of $Q_{TOA}$ to $c_{lw}$. The resulting linear relationship provided a value of 0.325 g.kg$^{-1}$ for $c_{lw}$ to obtain an increase of 2.2 W.m$^{-2}$ in $Q_{TOA}$. Thus we ran a second pre-industrial fully-coupled simulation of 500 years with this tuning. This experiment was interrupted when two of the bugs mentioned above were identified and fixed.

A second set of tuning was required as the bug correction altered the global 2-meter temperature. After several adjustments, the target of +2.2 W.m$^{-2}$ at TOA was expected to be obtained by setting $c_{lw}$ at 0.343 g.kg$^{-1}$. A ultimate fully-coupled experiment (called here PREIND) including this tuning was then branched on the previous simulation and run for 2,800 years.





## 3.2 Energetic and freshwater balances, equilibrium and drifts

**Table 2.** Global yearly averages of the radiative budget terms for PREIND last 100 years and historical experiment for the 1980–1999 period. Estimated values come from Trenberth et al. (2009) retrieved from Voldoire et al. (2013).

|  | PREIND | HISTORICAL |
|---|---|---|
| Net solar radiation at TOA | 239.9 | 242.7 |
| Outgoing LW radiation at TOA | 239.8 | 242.1 |
| $Q_{TOA}$ | 0.029 | 0.537 |
| Net solar radiation at surface | 171.3 | 172.7 |
| $Q_{surf}$ | 0.023 | 0.58 |
| Net LW radiation at surface | -67.29 | -66.73 |
| Incoming SW at surface | 197.7 | 198.7 |
| Outgoing SW at surface | 26.38 | 25.98 |
| Incoming LW at surface | 326.5 | 332.6 |
| Outgoing LW at surface | 393.8 | 399.4 |
| Sensible heat flux | -22.98 | -22.65 |
| Latent heat flux | -81 | -82.76 |

As expected, the imposed tuning induces a decrease in CRF in our preindustrial run (-18.97 W.m$^{-2}$ before tuning; -16.80
W.m$^{-2}$ after tuning, see section 4.1 for more details about CRF). Table 2 shows the terms of the energetic balance averaged
over the last decade of simulation. After 2800 years of integration, preindustrial $Q_{TOA}$ and Qsurf are stabilized to values close
to zero (respectively 0.029 W.m-2 and 0.023 W.m$^{-2}$), showing an acceptable energetic imbalance and limited spurious energy
excess or deficit. Energy conservation is not fully insured neither in the NEMO nor in the LMDZ component of the model,
but the global numerical leak is small. The non-conservative term, expressed as the difference between $Q_{TOA}$ and the ocean
heat content (OHC) change $\frac{\partial OHC}{\partial t}$ (Hobbs et al., 2016), is small (0.094 W.m$^{-2}$ over the last 1000 years) compared to IPSL-
CM5A (0.18 W.m$^{-2}$), which might be an improvement linked to corrections of energy conservation between IPSL-CM5A
and IPSL-CM5A2 versions of LMDZ (see section 2.1). Figure 3a indicates that this imbalance is constant after ∼300 years of
preindustrial simulation, and that OHC stabilizes after ∼2000 years of simulation. This ensures that the model drift is small,
therefore validating the use of IPSL-CM5A2 for multi-millennial simulations.


It takes about 500 years for sea surface temperatures and 2-meter air to stabilize after PREIND initialization (Fig. 3b). They
converge to 17.5 °C and 13.2 °C, when globally averaged, respectively. We thus missed the 13.5 °C target by 0.3 °C which is
reasonable when considering the very simple procedure used. Such a difference is also small enough compared to the regional
biases in sea surface temperature. An additional 1,500 years is required for the global average ocean temperature to stabilize,
as deeper ocean temperatures still cool by ∼0.1 °C to 0.2 °C during this period (Fig. 3c).



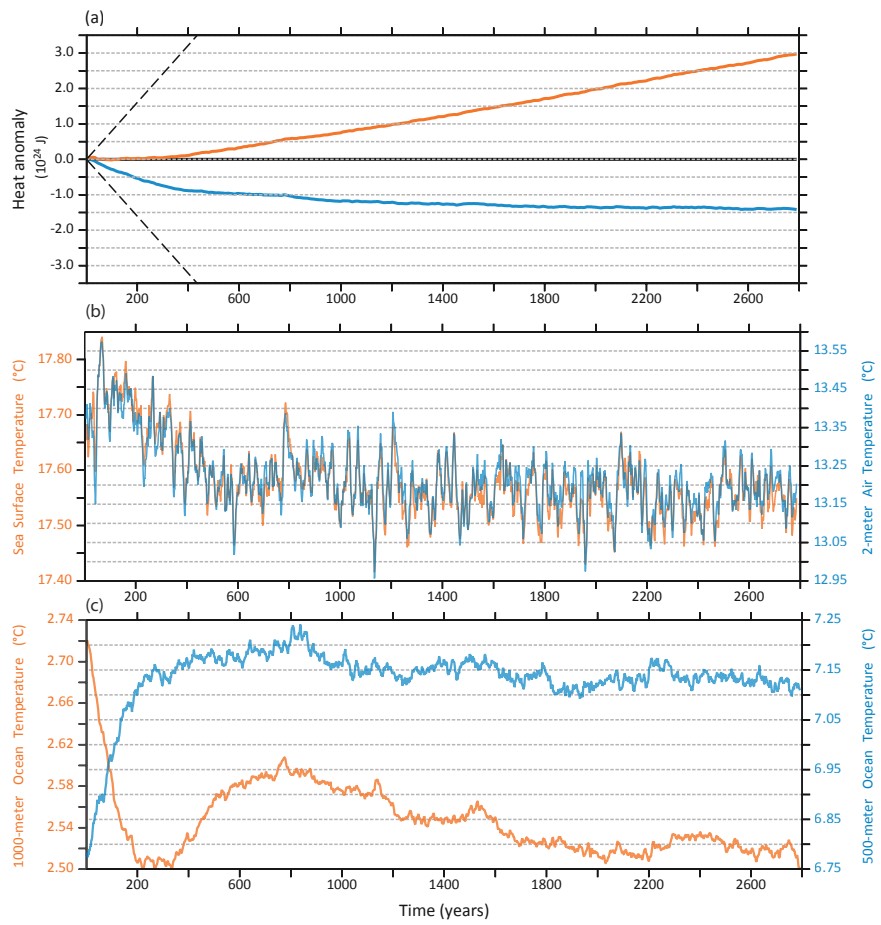

**Figure 3.** Time-series of globally averaged variables over the 2,800 years of the PREIND experiment. a) annual-mean heat content anomaly ($10^{24}$ J) implied by $Q_{TOA}$ (orange line, cumulative values) and OHC (blue line). As in (Hobbs et al., 2016), dashed black lines show the heat content implied by a $Q_{TOA}$ of $\pm\,0.5$ W.m$^{-2}$, the observed late-twentieth-century $Q_{TOA}$ (Roemmich et al., 2015). b) Sea-surface temperature (orange) and 2-meter air temperature (blue). c) 1000-m (orange) and 500-m (blue) ocean potential temperature.



Over the entire PREIND experiment, sea-surface height (SSH) depicts a constant positive drift of 0.19 m / century, revealing that the freshwater budget is not fully closed (not shown). This issue already existed in previous versions of the model, and is partly linked to the absence of a proper freshwater conservation between ocean and sea-ice components of NEMO. As a consequence, a linear decrease in global ocean salinity is simulated, corresponding to a steady global ocean freshening of $2 \times 10^{-3}$ psu / century. It appears that this drift varies with global sea-ice volume, as shown by abrupt 4xCO2 simulations that are characterized by a drift in SSH reduced to 0.01 m / century and a corresponding weaker freshening of the global ocean ($4.7 \times 10^{-4}$ psu / century, not shown). Such a drift might be problematic in the future if IPSL-CM5A2 is to be used for very long (transient-like) integrations. Improvements on this concern are expected from the future replacement of LIM2 by the more recent LIM3 version (Rousset et al., 2015), for which conservation is insured.

## 4    Comparison of IPSL-CM5A and IPSL-CM5A2 with observations

An historical experiment was performed starting from the last year of the 2800-year PREIND spin-up. This transient simulation was integrated for 155 years between 1850 and 2005. Boundary conditions are described in (Dufresne et al., 2013). Yearly evolution of long-lived greenhouse gases concentrations and total solar irradiance (TSI) correspond to the historical requirements of the CMIP5 project, with notably reduced TSI to consider the volcanic radiative forcing based on an updated version of Sato et al. (1993). Tropospheric aerosols, tropospheric and stratospheric ozone were prepared as described in (Szopa et al., 2013) and land use changes processed from crop and pasture datasets developed by Hurtt et al. (2011), as described in (Dufresne et al., 2013). Unless otherwise expressed, comparison between the historical run and observations is made by using a mean seasonal cycle for the period 1980-2005. The comparison between simulated and observed cloud radiative forcing is made using the Earth's Radiant Energy System (CERES) Energy Balanced and Filled Fluxes (EBAF) gridded dataset over the period 2000-2004 (Loeb et al., 2009).

### 4.1    Atmosphere mean state and identified biases

#### 4.1.1    Cloud radiative forcing

LMDZ (and hereby IPSL-CM5A) has been shown to underestimate low and mid-level cloud cover and overestimate high-level clouds at mid and high latitudes (Hourdin et al., 2013), biases shared with many climate models (Zhang et al., 2005). LMDZ cloud representation has been since strongly improved with the new set of physical parameterizations and associated increase in vertical resolution (Hourdin et al., 2013) that is currently being used for CMIP6 exercise. In IPSL-CM5A2, the biases of the standard parameterization translate into CRF, with an overestimation of shortwave CRF at mid and high latitudes (+15 to +25 W.m$^{-2}$), and an underestimation of outgoing longwave trapping in some convective regions of the tropics (-5 to -15 W.m$^{-2}$), when compared to the CERES dataset. The tuning depicted earlier leads to a global decrease in total CRF at all latitudes compared to IPSL-CM5A (Fig. 4).

The CRF strong overestimation in the southern hemisphere mid-latitudes is only slightly corrected, as less negative LW CRF



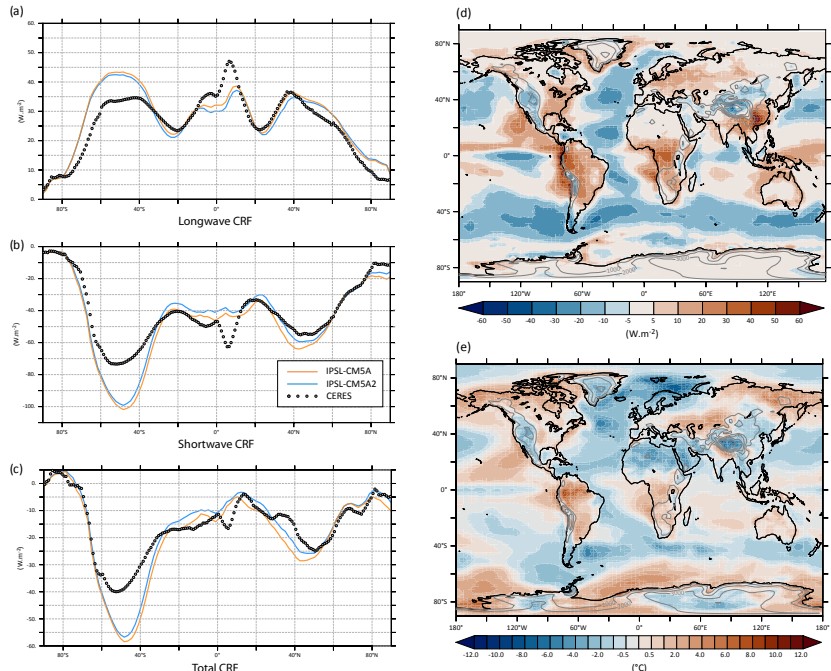

**Figure 4.** Zonally-averaged long-wave (a), shortwave (b) and total (c) cloud radiative forcing from IPSL-CM5A2 and IPSL-CM5A historical simulations, compared to the Clouds and the Earth's Radiant Energy System (CERES) gridded dataset (Loeb et al., 2009). (d). Yearly-averaged total CRF anomaly between IPSL-CM5A2 and CERES data. As CRF is a negative value, positive (negative) anomaly indicates an underestimation (overestimation) of CRF in the model. e) Yearly-averaged difference in near-surface air temperature (°C) between IPSL-CM5A2 and ERA-Interim (Dee et al., 2011). Grey contours indicate surface elevation.

and more positive SW CRF almost compensate (Fig. 4abc). In the southern tropics, the tuning mostly worsens the model biases in SW and LW CRF, in relation with a feedback loop in the continental tropical areas (namely the Amazon), where

convection is reduced. Tuning still improves total CRF between 15°N and 40°N, driven by the better estimation of SW CRF over the oceans. The overestimation of CRF over the eastern boundaries of Atlantic and Pacific oceans remain unchanged, and is likely related to a bad representation of high cloud decks over these regions (Fig. 4d). These biases translate into surface air temperature biases (Fig. 4e). Regions with an underestimated CRF (e.g. the eastern boundaries of the Pacific and the south Atlantic) are associated with up to 2.5 °C warm biases. Conversely, regions where CRF is overestimated, like the northern and

southern Atlantic and several mountain ranges (the Andes, the Rockies and the Tibetan Plateau) show cold biases (down to -6 °C for the Tibetan Plateau).





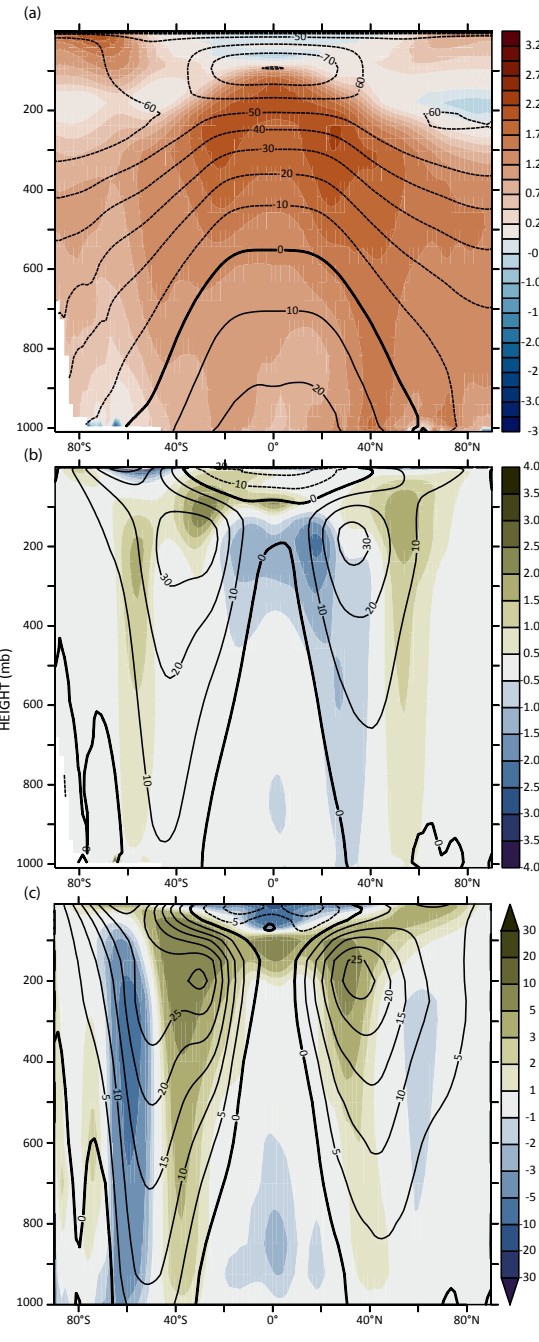

**Figure 5.** Zonally-averaged annual anomalies between: a) IPSL-CM5A2 and IPSL-CM5A temperature, b) IPSL-CM5A2 and IPSL-CM5A zonal wind, c) IPSL-CM5A2 and ERA-Interim (Dee et al., 2011) zonal wind. In each plot, contour lines indicate absolute values for IPSL-CM5A2, IPSL-CM5A2 and ERA-Interim, respectively.





### 4.1.2 Temperature and general circulation

The tuning of IPSL-CM5A2 induces a low latitude warming whose amplitude varies with height, from +1°C at 850 hPa to +2.5 °C at 200 hPa, when compared to IPSL-CM5A, as expected from the adjustment of the moist adiabatic lapse rate (Fig. 5a). As

a consequence of the increased the latitudinal temperature gradient between 200 and 300 hPa, the zonal jets expand towards high latitudes, especially in the Northern Hemisphere (Fig. 5b). This is an improvement, as IPSL-CM5A had been shown to underestimate the jets latitudinal extension (Barnes and Polvani, 2013). Despite this improvement, IPSL-CM5A2 still underestimates the jet extension in the Southern Hemisphere, as shown by a negative anomaly in zonal wind in the high latitudes and an overestimation of the zonal winds in the low latitudes. At 850 hPa and surface, IPSL-CM5A2 provides a better representation

of the zonal winds in the high latitudes: the negative anomaly with ERA-Interim is reduced along the Antarctic, showing that the westerlies have moved southward. The 850 hPa temperature anomalies also show that tuning has been effective in IPSL-CM5A2 to reduce the cold bias of the model, but also that the warm bias over the eastern tropical Pacific has strengthened. In the tropics, IPSL-CM5A2 depicts biases similar to IPSL-CM5A, i.e. over the Pacific Ocean, surface easterlies bad seasonal cycle lead to an overall yearly misposition, in relation with the double Intertropical Convergence Zone (ITCZ), and over the

Atlantic, easterlies strength is systematically underestimated. IPSL-CM5A2 also shows overestimated winter westerlies in the northern mid-latitudes, that translate into an anomalous wind stress over the northern Atlantic.

Sea-level pressure (slp) and surface wind patterns simulated by IPSL-CM5A2 are compared to ERA-Interim reanalysis in Fig. 6. The major circulation structures, such as the subtropical Highs of the Pacific and Atlantic, the December-January-

February (DJF) Atlantic mid-latitude Low, the June-July-August (JJA) easterlies reversal in the monsoon regions, are well-captured by the model. DJF westerlies are overestimated over the Atlantic. Conversely, the model reproduces well the slp patterns at low latitudes in JJA. Simulated slp values for JJA are also underestimated between 40°S and 60°S and overestimated along the Antarctic coastline, in line with the zonal wind bias depicted earlier. As a consequence, westerlies velocity maxima are restricted to 45°S instead of 50°S in the reanalysis (Fig. 6cf). The southern eastern Pacific is an exception to this general

pattern, as sea level pressures are overestimated between 45°S and the Antarctic coastline over this region, associated with very low surface winds velocity values. Over western Africa, the monsoonal surface winds intensity and position are well-simulated, in line with the good positioning of the Sahara Thermal Low (Fig. 6de). Both winter and summer Asian monsoon surface wind patterns are correctly positioned as well, but their intensity is underestimated. Overall, the model overestimates the strength of surface easterlies, both in winter and summer. This is particularly visible over the Sahara region in DJF (Fig. 6ab) and the

northern part of South America in JJA (Fig. 6de).

### 4.1.3 Tropical precipitation

Regarding precipitation, IPSL-CM5A2 and IPSLCM5A essentially share the same qualities and biases, i.e. an underestimation of rainfall in the mid-latitudes (20°-40°) of both hemispheres, an overestimation of rainfall in the southern tropics, characterized on the yearly zonal average by a peak at ∼10 °S corresponding to a "double ITCZ" 7e. Dry biases (-0.5 to -2 mm.d$^{-1}$) are



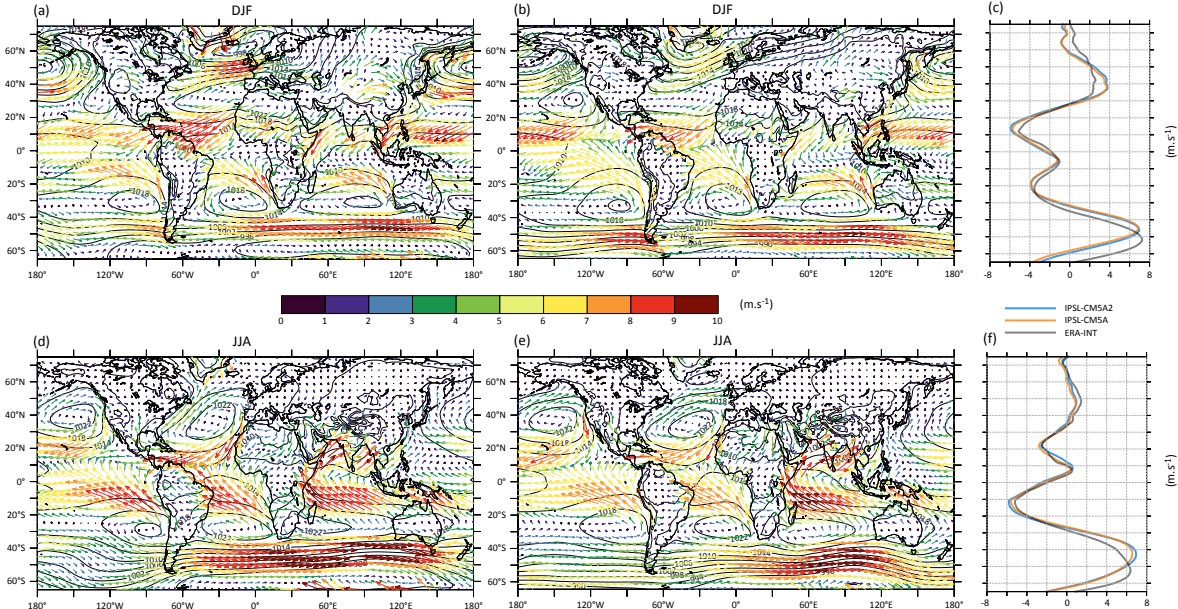

**Figure 6.** Mean sea-level pressure and near-surface wind velocity (colors) and direction for IPSL-CM5A2 (a, d) and ERA-Interim (b, e). c, f. Zonally averaged near-surface wind velocity (m.s$^{-1}$) for IPSL-CM5A, IPSL-CM5A2 and ERA-Interim. Positive (negative)values are eastward (westward).

marked over the western edges of the northern and southern Atlantic at mid-latitudes, as well as the northwestern Pacific in JJA. The regional anomaly between IPSL-CM5A2 and GPCP data also show that the tropical biases vary with longitude: namely rainfall is overestimated over the equatorial western Indian, equatorial Africa and south equatorial Atlantic (+1 to +3 mm.d$^{-1}$) in DJF (Fig. 7ac), whereas there is a strong underestimation of tropical rainfall over South America, specifically over the Amazon basin (down to – 5 mm.d$^{-1}$).


   **South America.** The dry bias over South America and every regions of the Amazon basin (Fig. 8) is a long-standing feature of the IPSL coupled models that likely has multiple origins, as precipitation over this region depends both on moisture advection from the Atlantic (and thus from the capability of the model to reproduce correct atmospheric dynamics and SSTs) and continent recycling of water through evapotranspiration. Dufresne et al. (2013) showed that soil depth had to be increased

from 2 to 4 meters in IPSL-CM5A to favor seasonal water retention in the soil, allow vegetation to grow, and thereby reduce a too strong dry bias over the Amazon. We kept this tuning in IPSL-CM5A2. Still, the seasonal 850 hPa wind divergence shows that a strong dipole between restricted convective zones (Fig. A4) and expanded subsiding zones is simulated over tropical South America. This pattern induces restricted areas with strong (>8 mm.d$^{-1}$) convective precipitation (northwestern Amazon in JJA, south-central Amazon in DJF) connected to larger zones where rainfall is limited by subsidence. The increase in

the dry bias over South America between IPSL-CM5A and IPSL-CM5A2 results from a feedback loop linked to vegetation



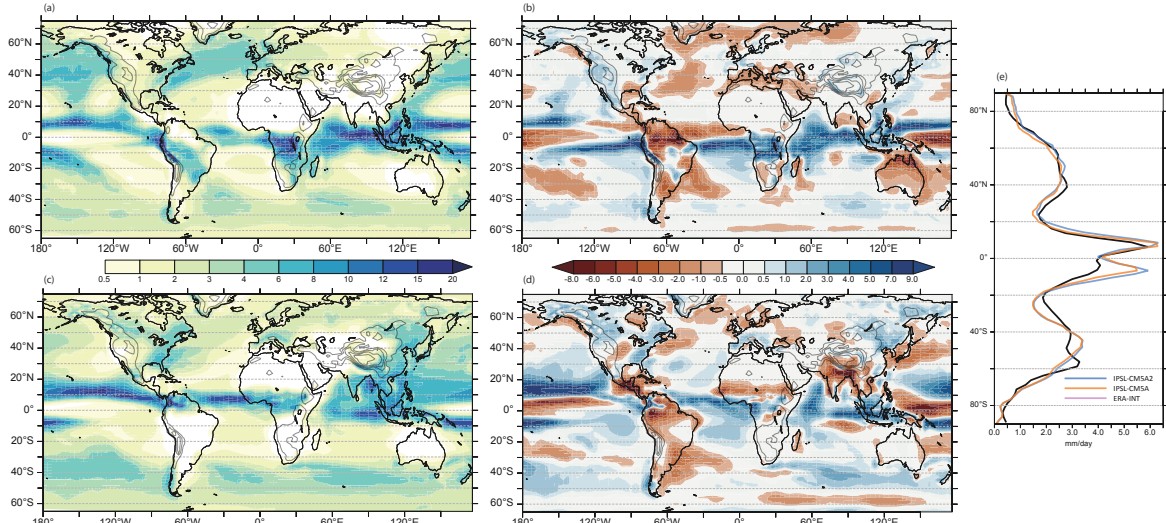

**Figure 7.** Seasonally-averaged precipitation (mm/day) for IPSL-CM5A2 (a,c) and anomaly between IPSL-CM5A2 and GPCP data (b, d). Top is DJF, bottom is JJA. e) Zonal average of absolute annually-averaged precipitation (mm/day) for IPSL-CM5A2, IPSL-CM5A and GPCP data.

change. The tuning of the clouds produces excessive warming (Fig. 4), especially over the northwestern Amazon basin, that combines with DJF subsidence to strongly reduce rainfall amounts and thereby tropical rainforest cover (not shown). In line with earlier deforestation experiments carried out with the IPSL model (Davin and de Noblet-Ducoudré, 2010), such a decrease in tree cover has a strong feedback on temperature, as it leads to a decrease of latent heat flux and increase in sensible heat
fluxes (overpassing the albedo increase, that tends to cool surface air temperature). As an illustration, the 2-meter temperature anomaly over northwestern Amazonia between IPSL-CM5A2 and IPSL-CM5A reaches + 3.5°C, with IPSL-CM5A2 annual mean two-meter temperature reaching 29.9 °C (not shown). Vegetation decrease also further reduces water recycling through lower evapotranspiration as confirmed by a supplemental historical experiment with IPSL-CM5A2 initialized with observed values of vegetation rather than the results from PREIND that shows that prescribing rainforest over the Amazon basin cools
two-meter temperature by 1 to 3 °C and increase annual rainfall amount by 2 mm.d$^{-1}$ (not shown). The seasonal cycles of precipitation computed over the northern and southern parts of the Amazon basin show the simulated rainfall collapse during the respective hemispheric winters, while the year-round underestimation of rainfall in the western part of the basin seems to related to a mislocation of rainfall, that occur on top of the Andes instead of the foothills.

**Africa**. Conversely, precipitation rates are overestimated over equatorial Africa (Fig. 7). The divergence of 850 hPa wind shows
that the model overestimates convergence of moist air masses south of the equator in DJF (Fig. A4). Moreover the +0.5 to +2 °C anomalies in the eastern tropical Atlantic SST likely favor enhanced evaporation and moisture advection over the continent. Over central Africa (Fig. 15), rainfall seasonal cycle is bimodal with two humid seasons, but the maxima lag by one month when compared to data. More importantly, rainfall amount are largely overestimated by $\sim +66$ % during the humid seasons.



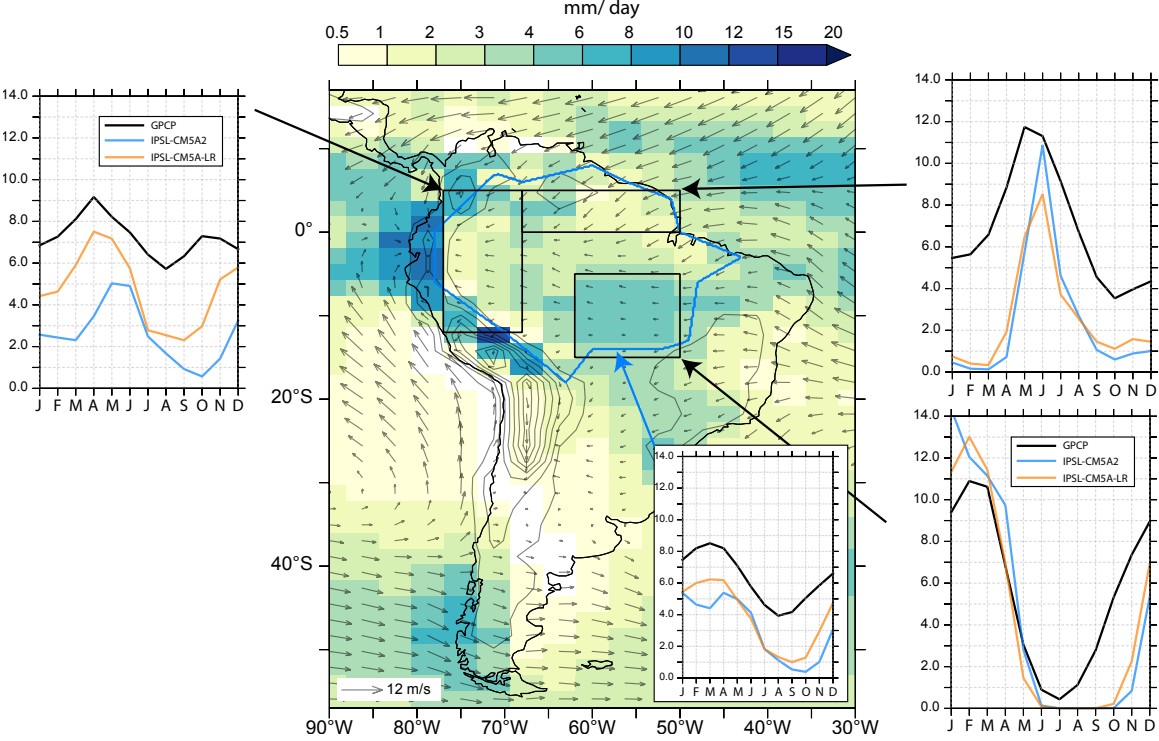

**Figure 8.** Rainfall patterns over South America. The map shows the precipitation annual mean (mm/day) simulated in IPSL-CM5A2 historical experiment. Insert on the map shows the seasonal cycle over the entire Amazon basin (blue line) for IPSL-CM5A, IPSL-CM5A2 and GPCP data. Inserts at the edges of the map show the seasonal cycles for three subregions of the Amazon basins.

Over the western African monsoon region, IPSL-CM5A and CM5A2 both lag the start of the rainy season, leading to an overall
underestimation of rainfall before the monsoon peak. However this rainfall maximum occurs in August as for observations, and the subsequent decrease in simulated correctly.

**Indian monsoon and Indonesia** . Compared to IPSL-CM5A, the Indian monsoon has been improved in IPSL-CM5A2 (Fig. 10ab). The 1-month lag that characterized IPSL-CM5A is still present, but the amplitude of the seasonal cycle of precipitation, although still underestimated, has been enhanced, peaking at 7.3 mm.d$^{-1}$ in August (5.6 mm.d$^{-1}$ in August in IPSL-CM5A),
closer to observations (9.3 mm.d$^{-1}$ in July in GPCP data). This improvement is linked both to higher large-scale and convective precipitation during summer, consecutive of the enhanced hydrological cycle linked to the tuning-induced warming. However, the artificially high precipitation rates on the Himalayan foothills have not been corrected. They are made mostly of large-scale precipitation that are overestimated due to a too strong cooling over the highest altitude grid points.

Fig. 10c shows Hovmöller diagrams for GPCP dataset compared to IPSL-CM5A2. Although at the first order, the seasonal cy-
cle of precipitation is correctly represented over the "Maritime Continent", IPSL-CM5A2 depicts a strong humid bias over the



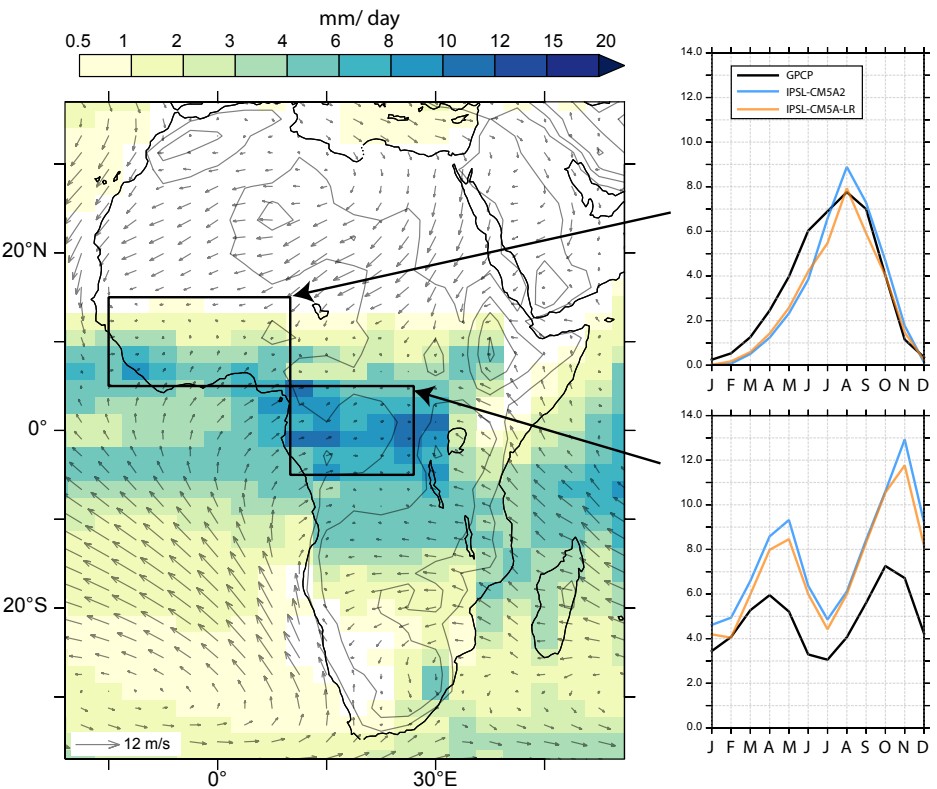

**Figure 9.** Rainfall patterns over Africa. The map shows the precipitation annual mean (mm/day) simulated in IPSL-CM5A2 historical experiment. Inserts on the sides of the map show the seasonal cycle over the Congo basin and the west African monsoon region for IPSL-CM5A, IPSL-CM5A2 and GPCP data.

region (up to +5 mm.d$^{-1}$ during the humid season). This bias has been described earlier with LMDZ pre-industrial simulations run either in atmospheric-only or fully-coupled mode (Toh et al., 2018; Sarr et al., 2019) and it was shown that LMDZ systematically overestimates rainfall over both land and sea surfaces of the Maritime Continent, together with western Indian Ocean and Bengal Bay. Given the complexity of the Indonesian region, with a myriad of islands that the model spatial resolution cannot capture, it is no surprise that rainfall are misrepresented in this model (Qian, 2008).



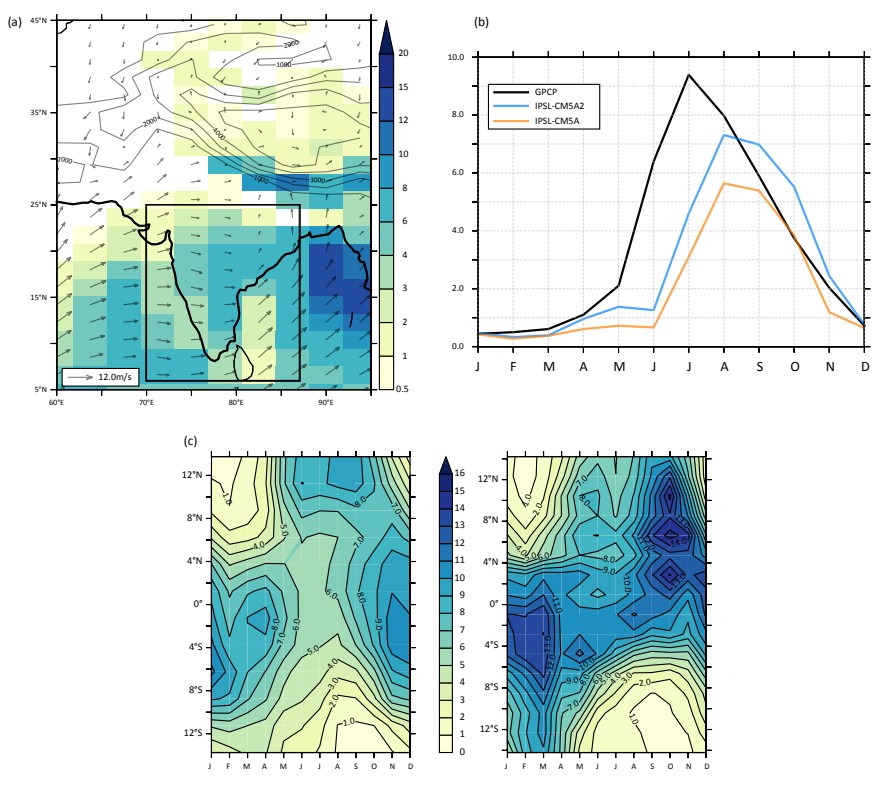

**Figure 10.** Hovmöller diagrams of precipitation (mm.d$^{-1}$) averaged over a 95°E-115°E range. Left, GPCP, right, IPSL-CM5A2.

## 4.2 Ocean mean state and identified biases

The annual SST averaged between 50°S-50°N simulated by IPSL-CM5A2 for the last decades of the historical run is 22.3 °C
(1 °C warmer than IPSL-CM5A), a value in the higher range of CMIP5 ESMs and consistent with observations (Fig. A3).
The warm bias over eastern subtropical Pacific and Atlantic oceans is more marked in IPSL-CM5A2 than in IPSL-CM5A as

a response to CRF tuning (Fig. 11ab). This bias has already been reported in several coupled models using NEMO, namely
CNRM-ESM1 (Séférian et al., 2016), CNRM-CM5.1 (Voldoire et al., 2013), IPSL-CM4 (Marti et al., 2010) and IPSL-CM5A
(Dufresne et al., 2013). As stated in Voldoire et al. (2013), ocean-only experiments (Griffies et al., 2009) have suggested that
such a warm bias was likely linked to "poorly resolved coastal upwellings and underestimated associated westward mass
transport due to the coarse model grid resolution". The 0.5-2 °C warm bias over the Antarctic circumpolar current (ACC) is

comparable to IPSL-CM5A and can be related to the poor simulation of the CRF over this region (Hyder et al., 2018) and
the associated mispositioning of mid-latitude westerlies depicted above. Conversely, the tuning has reduced the cold bias in
the Pacific and Atlantic gyres. Surface salinity biases are rather similar between both models. The surface salinity anomaly
follows the E-P pattern over the tropical oceans (not shown), showing the influence of the double ITCZ issue on the tropical



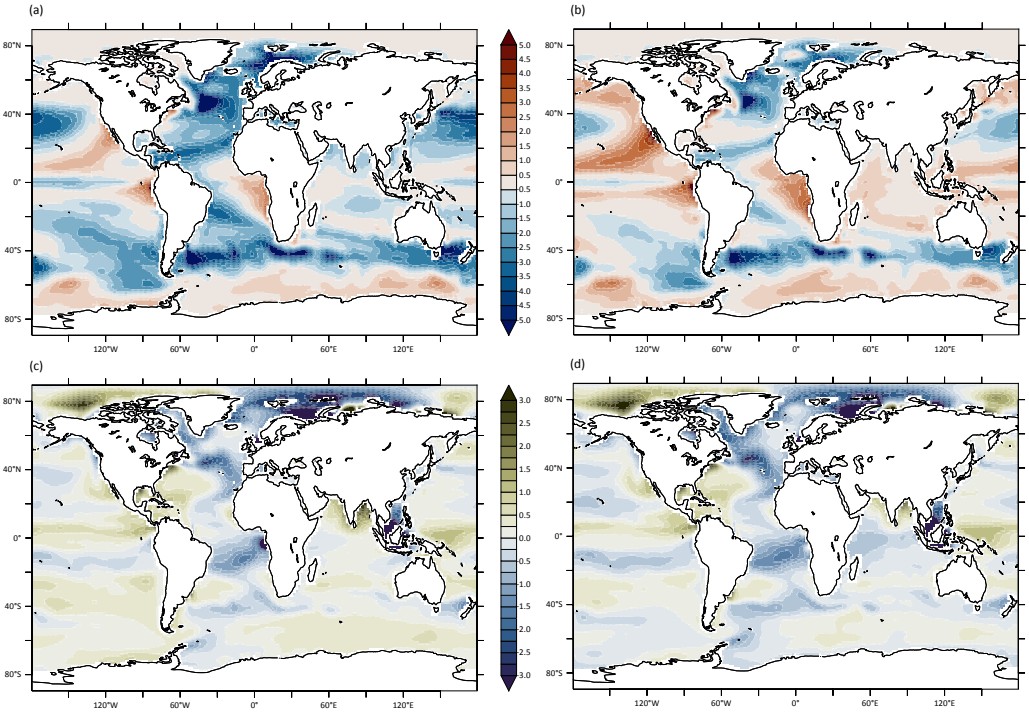

**Figure 11.** Anomalies of surface temperature (top), and surface salinity (low), between IPSL-CM5A and data (left) and IPSL-CM5A2 and data (right).

ocean freshwater balance. Both models depict too fresh surface waters in the northern Atlantic and locations of deep water
formation (Fig. 11cd). The ocean also depicts temperature biases at depth both in the Atlantic and Pacific oceans (Fig. 12).
Zonally-averaged potential temperature anomalies show that the warm bias in the northern Atlantic between 1000 and 3500 m
has been amplified, as well as the warm bias in the northern Pacific, that concerns the entire water column in IPSL-CM5A2.
The strong ($> +2.5$ °C) overestimation of temperatures in the subsurface waters of the southern hemisphere is almost identical
in both versions of the model. The cold ($\sim -1$ °C) bias in the high latitudes of the southern hemisphere has been slightly
reduced, but is still present due to the lack of strong enough overturning in this region. Sea-ice extent (Fig. 13) has been
improved in the North Atlantic sector, but remains overestimated. The Nordic Seas are covered by sea ice in winter in IPSL-
CM5A, while IPSL-CM5A2 shows overestimated sea ice mostly in the Barents Sea. In the Labrador Sea, both IPSL-CM5A
and IPSL-CM5A2 have overestimated sea ice, likely preventing water mass sinking and convection in this region. In contrast,
sea ice is underestimated in both versions for the North Pacific sector, i.e. in the Bering and Okhotsk seas.

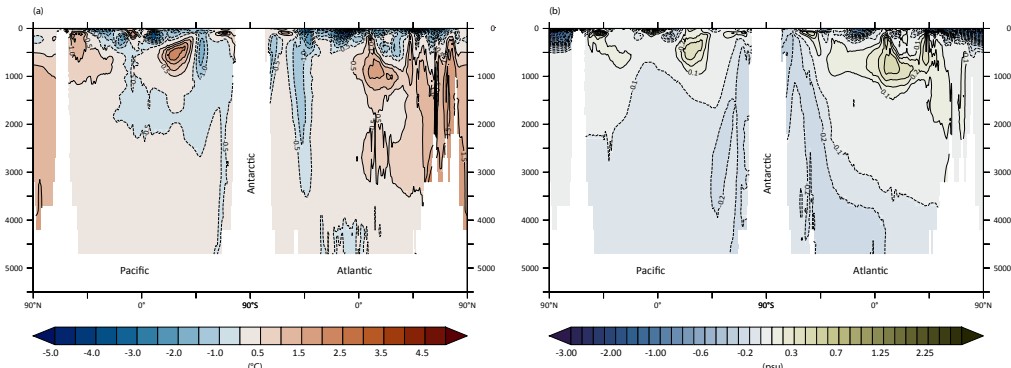

**Figure 12.** Yearly-averaged anomalies between IPSL-CM5A2 temperature (a) and salinity (b) averaged over 1980-1999 and the WOA2013 observations (Locarnini et al., 2013; Zweng et al., 2013).

### 420  4.2.1  Ocean Heat Transport and Meridional Overturning Circulation

IPSL-CM5A2 ocean meridional heat transport is higher (in absolute value) than the one simulated with IPSL-CM5A in both hemispheres (Fig. 14). This increase in the Northern Hemisphere is mainly related to the increase in the Atlantic meridional heat transport in the mid-latitudes (not shown). When compared to observations, the two model versions have asymmetric biases, with the northward transport being underestimated in the Northern Hemisphere and the southward transport overestimated in

the Southern Hemisphere low latitudes (between the equator and 35°S). Southward heat transport is too strong between the equator and 40°S and is in better agreement with data than IPSL-CM5A between 40°S and 60°S. The Atlantic Meridional Overturning Circulation (AMOC) is more vigorous in IPSL-CM5A2, with a maximum in depth (below 500 m) and latitude (from 30°S to 60°N) in preindustrial conditions moving from $10.3 \pm 1.2$ Sv (Escudier et al., 2013) to $12.02 \pm 1.1$ Sv (taken for the last 1000 years of simulation, Fig. 15a). The last 20 years of the 20th century in the historical run depicts a less vigorous

circulation in the Atlantic, with a mean value for AMOC index of $10.5 \pm 0.9$ Sv. The convection sites have been slightly modified between the two versions of the model. While the strongest convection was located in the North Atlantic subpolar gyre (SPG), south of Iceland, in the IPSL-CM5A version (Escudier et al., 2013), in IPSL-CM5A2, the Greenland sea shows the deepest mixed layer depth, which is larger than 1500 meters in winter (Fig. 13cd). Nevertheless, there is still a very active deep convection site south of Iceland, which is not realistic. The strengthening of the Nordic Seas convection can be related with the

decrease of its sea-ice cover in winter (Fig. 13). As a consequence, denser water is now produced in this site, which improves the overflow through Greenland-Iceland-Scotland straits. Indeed, the water denser than $27.8 \, \mathrm{kg.m^{-3}}$ flowing southward reaches 1.5 Sv in IPSL-CM5A2, while it was only 0.2 Sv in IPSL-CM5A. It remains largely underestimated as the observations from (Olsen et al., 2008) indicates around 6 Sv of overflow in total from observation-based estimates over the last few decades. Besides, the poleward shift of the westerlies in the North Atlantic stimulate a northward shift of the boundary between the

Atlantic subtropical and subpolar gyres, as shown by the barotropic streamfunction anomaly (Fig. 15b). This may contribute to bringing more salt into the subpolar gyre, which may also act to intensify the AMOC in IPSL-CM5A2.



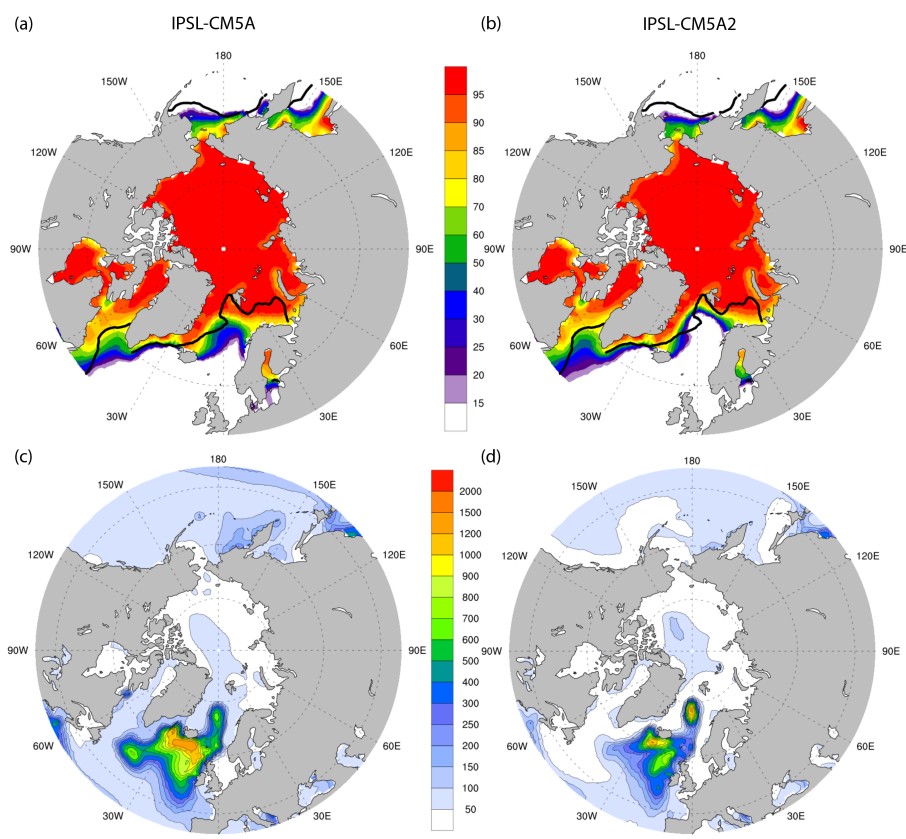

**Figure 13.** a,b. March sea-ice cover (%) for IPSL-CM5A (left) and IPSL-CM5A2 (right) historical runs. Black contour indicates the 15% sea-ice cover interval in observations. c,d. January-February-March average of mixed-layer depth (meters) for IPSL-CM5A (left) and IPSL-CM5A2 (right).

| | Reference for observations | Observations | IPSL-CM5A2 | IPSL-CM5A | CNRM-CM5.1 (Voldoire et al., 2013) |
|---|---|---|---|---|---|
| **Bering Strait** | (Woodgate et al., 2005) | 0.8 | 1.2 | 1.2 | 1.4 |
| **Indonesian throughflow** | (Sprintall et al., 2009) | 15.0 | 12.6 | 17.2 | 11.3 |
| **Drake passage** | (Cunningham et al., 2003) | 136.7 | 101.9 | 100.3 | 87.2 |
| **Florida strait** | (Baringer and Larsen, 2001) | 31.75 | 19.6 | 20.2 | 27.4 |

**Table 3.** Mass fluxes through major ocean gateways, in Sverdrups.

### 4.2.2 Fluxes in the major ocean gateways

We computed oceanic mass transport through selected longitudinal and latitudinal cross-sections corresponding to the major gateways (Table 3), and compared them to IPSL-CM5A, estimates from observations, and values simulated by CNRM-CM5.1,



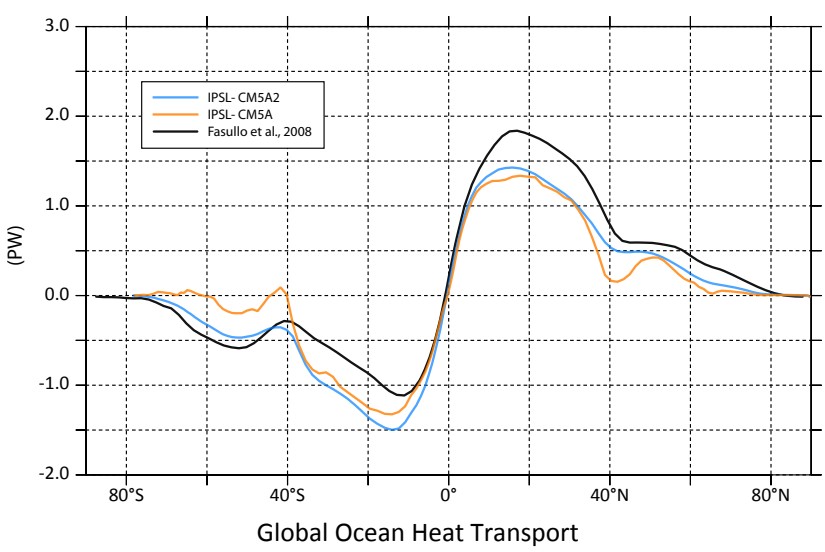

**Figure 14.** Global meridional heat transport in the ocean, in petawatts (PW).

that uses NEMO as well (Voldoire et al., 2013). These diagnostics highlight that the Antarctica Circumpolar Current (ACC) is
underestimated by IPSL-CM5A2 by around 25%, as for the other configuration considered here, whereas despite the absence
of high resolution and explicit resolution of Indonesian gateways, the overall Indonesian throughflow tends to be close to
observation-based estimate (underestimation of 16%). Compared to observations and CNRM-CM5.1, the flow in the Florida
strait is quite weak (weaker than observation by 37%), in line with the simulated "sluggish" AMOC. The increase of the AMOC

from IPSL-CM5A to IPSL-CM5A2 has not been enough to strengthen this current, also largely resulting from wind forcing.
Also, the absence of resolved eddies can also play a significant role in this underestimation.

### 4.2.3   Net Primary Production (NPP)

To illustrate some of the capabilities of IPSL-CM5A2, we also evaluate the simulated oceanic NPP (Fig. 16). The global
integrated NPP over the historical period (here averaged over 1980-1999 from the historical simulation) is 47.5 PgC.y$^{-1}$,

which is a strong increase when compared to 30.9 PgC.y$^{-1}$ simulated for IPSL-CM5A (Bopp et al., 2013), making IPSL-
CM5A2 being closer to the 52.1 PgC.y$^{-1}$ global estimate based on SeaWifs remote-sensing observations (Behrenfeld and
Falkowski, 1997). The representation of NPP at the spatial scale compares well to observation-based estimates for the tropical
and Southern Hemisphere, but depicts large biases in the Northern Hemisphere, especially in the North Atlantic where NPP is
largely under-estimated (Fig. 16c).





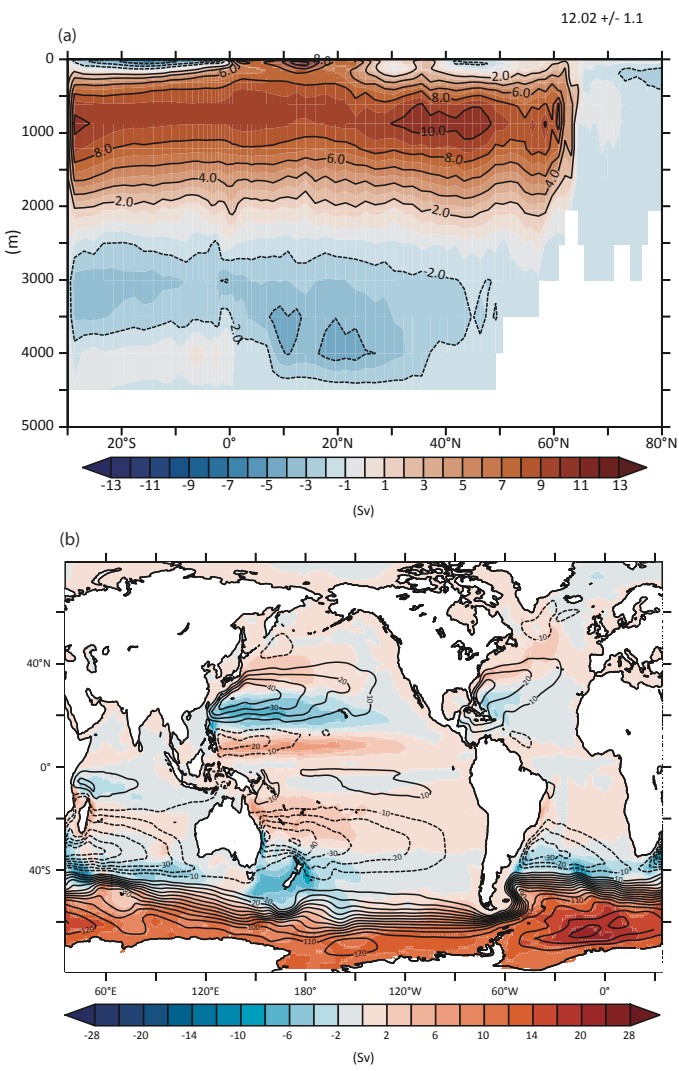

**Figure 15.** a. Atlantic meridional streamfunction (Sv) averaged over the 1980-1999 period of the historical run. b. Barotropic streamfunction (bsf) of IPSL-CM5A2 PREIND experiment (contour, in Sv) and difference between IPSL-CM5A2 and IPSL-CM5A PREIND experiments (color, in Sv).

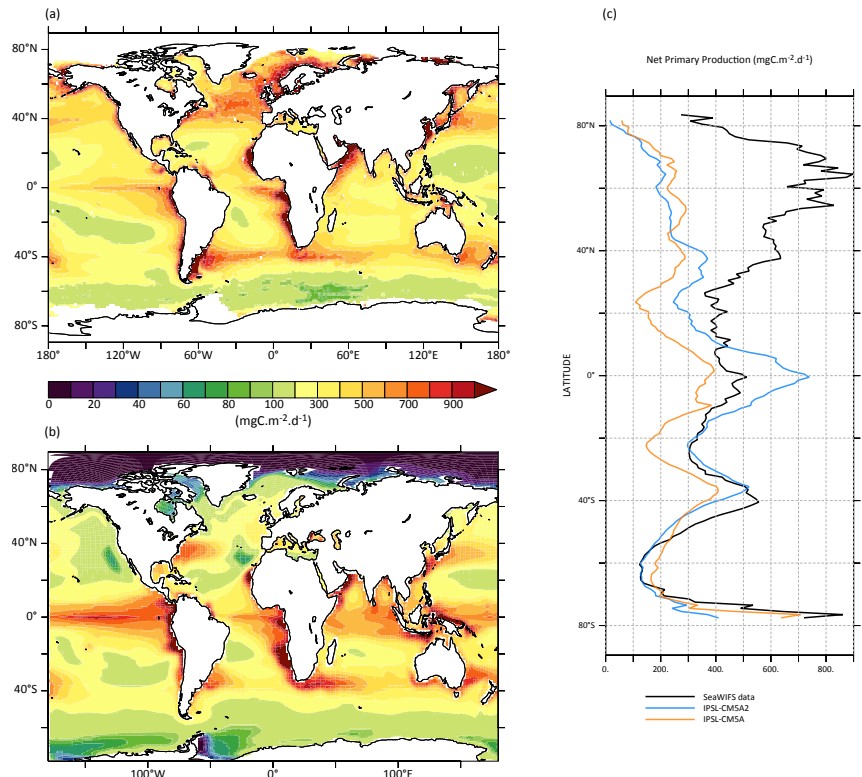

**Figure 16.** Mean vertically integrated NPP retrieved from SeaWifs observations (a), and simulated by IPSL-CM5A2 (b). c. Latitudinal variations of Mean vertically integrated NPP, zonally-averaged, for SeaWifs (black),IPSL-CM5A (orange) and IPSL-CM5A2 (blue).

### 4.2.4 Ocean variability

The main modes of variability of the AMOC are analyzed with an empirical orthogonal function (EOF) of the yearly Atlantic meridional overturning streamfunction (Fig. 17, top). The first EOF is a monopole in IPSL-CM5A and IPSL-CM5A2, but IPSL-CM5A2 shows less loading in the subpolar gyre between 40°N and 60°N. The associated principal component shows a significant 20-yr variability in the case of IPSL-CM5A. Such variability was previously linked to advection of surface salinity anomalies from the western boundary to the Eastern Atlantic (Escudier et al., 2013), and westward propagating planetary waves in subsurface (Ortega et al., 2015). However, the peak of variability at 20-yr timescale is not significant any more in IPSL-CM5A2 (see spectra in Fig. 17, center). This difference can be explained by the different tuning of the two model versions.

Gastineau et al. (2018) used IPSL-CM5A-MR, a variant of IPSL-CM5A with an improved horizontal resolution in the atmosphere (2.5° × 1.25° vs 3.75° × 1.87°), and also found that a warmer mean state led to a reduction of the 20-yr variability

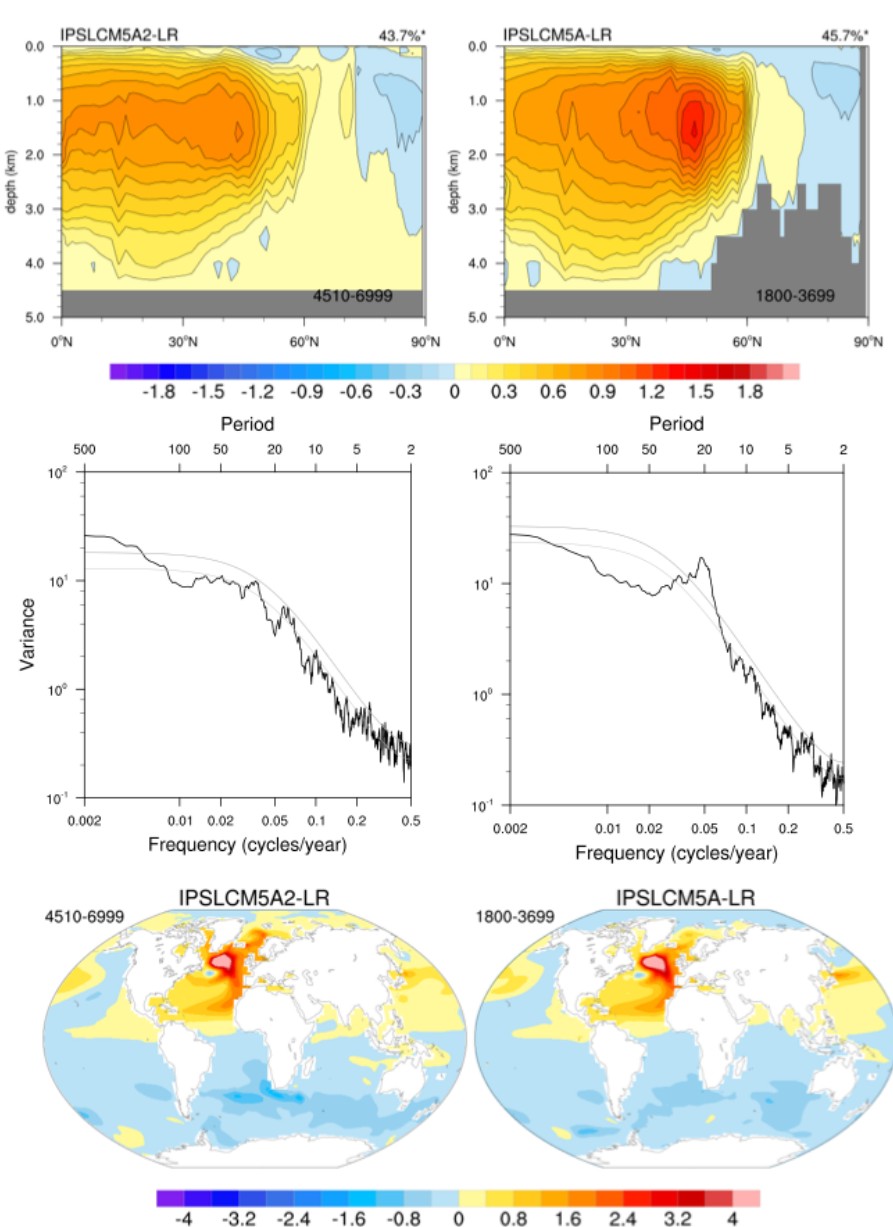

**Figure 17.** (Top) First EOF of the yearly AMOC, in Sv, (center) variance spectrum of the associated PC1 and (bottom) AMV pattern in K.





in IPSL-CM5A-MR. Such a warmer mean state modifies the subpolar gyre currents and the subsurface stratification in the Eastern Atlantic Ocean where the mixed layer is deepest in IPSL-CM5A models. These changes were suggested to explain a smaller growth of baroclinic instabilities triggering the propagation of westward propagating anomalies (Gastineau et al., 2018). It is therefore likely that the different tuning of IPSL-CM5A2 led to similar changes. The SST associated with the first
principal component of the AMOC also shows weaker SST anomalies in the subpolar gyre when compared to IPSL-CM5A (not shown). The Atlantic multidecadal variability (AMV), was calculated from the low-pass filtered (61-month running mean) North Atlantic SST in 0°N-60°N 80°W-0°E. The AMV pattern is given by the regression of the SST onto the AMV index. The AMV patterns are simimar in IPSL-CM5A and IPSL-CM5A2, with a slightly more intense subpolar gyre SST (6.8°C vs 6.2°C for the maximum value in the subpolar gyre) in IPSL-CM5A2. Larger positive SST anomalies in IPSL-CM5A2 are also
simulated in the Nordic Seas and Irminger sea associated with the AMV. The larger AMV in IPSL-CM5A2 is consistent with the smaller sea-ice cover in the North Atlantic, and a more intense signature of the atmospheric forcing.

The Pacific variability in IPSL-CM5A2 is identical to that found in IPSL-CM5A. The El Niño Southern Oscillation (ENSO) is characterized by the Niño3.4 time series. IPSL-CM5A and IPSL-CM5A2 show a similar power spectrum (Fig. A5) which
is comparable to observations (Bellenger et al., 2014). Both versions have the same bias in the seasonal phase locking (Fig. A5), as the monthly standard deviations are larger in May and June in CM5A and CM5A2, while the observed maximum is in January and December. The ENSO impacts are evaluated using spatial composites of the SST over ocean, 2-meter air temperature over land and sea level pressure. The composites are built using the Niño3.4 time series, and calculating the difference of all years where the Niño3.4 exceeds one standard deviation, minus years less than 1 standard deviation. Fig. A6
illustrates that the ENSO impacts are identical in both versions of the model, with a systematic bias in the position of the Pacific equatorial SLP anomalies which extend too much west toward the Pacific warm pool. However, the SLP anomalies display a realistic pattern in the Northern and Southern Hemisphere, with the associated Pacific North American and Pacific South American pattern, as in other CMIP5 models (Weare, 2013). The Pacific Decadal Oscillation (PDO) was also calculated as the first EOF of the monthly Pacific SST anomalies north of 20°N (Fig. A7), which illustrate a pattern with positive SST anomalies
over the equatorial Pacific, negative anomalies in the northwest Pacific, and horseshoe pattern of positive SST anomalies in the Northern Hemisphere. The IPSL-CM5A model generally simulates the characteristics of the observed PDO, as found by (Fleming and Anchukaitis, 2016) or (Nidheesh et al., 2017), as well as the associated anomalies in the Atlantic and ocean basins. The IPSL-CM5A2 version simulates a PDO nearly identical to that of IPSL-CM5A, so that the mean state changes have no impact on the PDO state.

## 5   IPSL-CM5A2 for deep-time paleoclimate modelling, a case-study

This section depicts both the technical developments and the choices to be made regarding boundary conditions for a typical deep-time simulation with IPSL-CM5A2. We describe the generation of a new grid for the ocean model for deep time applications and the generation of boundary conditions, component-wise. To illustrate our point, we run a 3000-year-long numerical

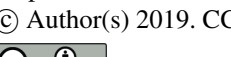


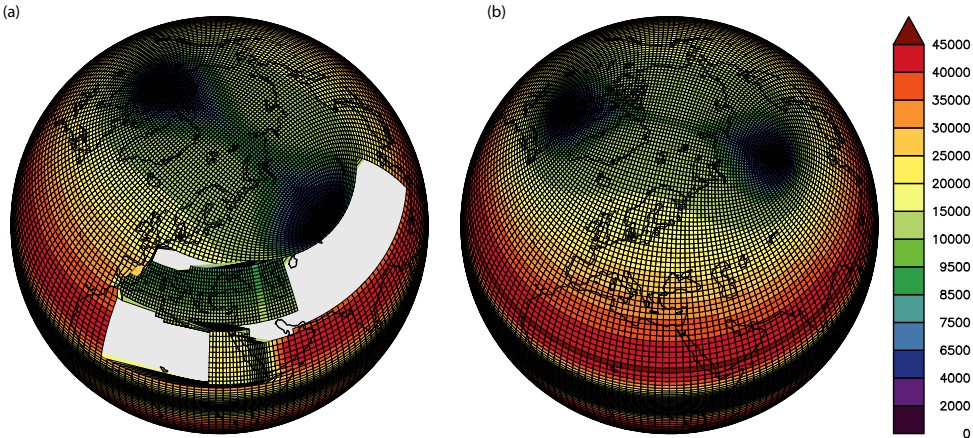

**Figure 18.** Comparisons of ORCA2 grid (left) and PALEORCA grid (right). Color shading shows spatial resolution (km2). Grey-shaded areas on ORCA surround the refinements of the grid over the Mediterranean, Red, Black and Caspian seas. The centers of blue areas show the two poles, that have been rotated in PALEORCA.

simulation of the Cretaceous as a case study. We present a very brief analysis of the model timeseries to illustrate the long ad-

justment required to reach equilibrium in deep-time simulations. As the focus of the paper is the description of IPSL-CM5A2, we only provide a very basic description of the simulated climate, whose characterization will be the focus of a subsequent study.

### 5.1 Generating a new grid for NEMO

In addition to the aforementioned computational performances, technical issues related to the oceanic component of the model

also have long prevented the use of any version of the IPSL ESM for deep-time climate modelling. The main lock was that NEMO runs on a tripolar grid (ORCA2, with two poles in the northern hemisphere and one over Antarctica) that was designed, amongst other objectives, to avoid any singularity points on the computational domain (i.e. the ocean domain) (Madec, 2015; Madec and Imbard, 1996). Specifically, north of 20 °N the mesh is made of a series of embedded ellipses, the foci of which are the two north poles of the grid that are by design placed over continents (Fig. 18a). Southward the grid is a Mercator (i.e.

identical longitudinal ant latitudinal grid spacing) grid, defined down to 78°S, except from local refinements applied in the tropics, so that the meridional resolution reaches 0.5° at the Equator and over the Red, Black, Caspian and Mediterranean Seas where resolution is about 1°.

Because the grid poles are located over North America, Asia and Antarctica, changing the land-sea mask to account for continental drift can shift the singularity points into the computation domain. In addition, a number of ocean routines include

hard-coded specifications that apply to some narrow modern straits, such as the Gibraltar strait. In these regions, the resolution of the grid is much larger than the width of the straits, thereby requiring the outflow of water to be prescribed. Although needed





to better simulate modern climates, these features are irrelevant for deep-time simulations. To overcome these issues, we design a new PALEORCA grid based on techniques and code designed by G. Madec and A. Sellar. The southern hemisphere (SH) pole is conserved, which restricts the application of IPSL-CM5A2 to the last 100 million-years (Antarctica has been located

over the South Pole since ca. mid-Cretaceous). In the northern hemisphere (NH), new ellipses are computed, changing the location of the two singularity points to eventually obtain a grid that can effectively be used for the last 100 million-years. To this end, and because the grid structure requires both NH poles to be diametrically opposed and limits the range of latitude in which the location of the poles can be chosen, the two poles are placed respectively at 106°E, 60°N and 74°W, 60°N (Fig. 18b). Above-mentioned refinements over the Mediterranean and Eurasian Seas are removed, but the tropical refinement is

kept. We also take advantage of the well-defined coding guidelines of NEMO to create a "PALEORCA" configuration within the available oceanic configurations of the model. In this configuration, no hard-coded water exchanges across the globe are prescribed.

## 5.2 Application: Generating boundary conditions for deep-time climate modelling

The Cretaceous is historically known for its warmer than present climatic state without permanent ice sheets at the poles. Here,

we present a simulation of the Turonian stage (circa 90 Ma), during which the apex of the Cretaceous warmth was reached. CO2 estimates reconstructed from proxy-based studies vary from 500 ppm to > 2000 ppm with large uncertainties (Wang et al., 2014). In our simulation, we impose an atmospheric $CO_2$ concentration of 1120 ppm (4x preindustrial values) and we remove polar ice sheets. In addition, we keep the modern orbital configuration as no numerical solution for orbital parameters exists for the Cretaceous (Laskar et al., 2004). The basis of our Turonian configuration is the Cenomanian/Turonian paleogeography

of (Sewall et al., 2007).

### 5.2.1 Oceanic boundary conditions

**Bathymetry**. The Turonian bathymetry (Fig. 19b) is obtained from the merging of two datasets. First, the regular longitude-latitude Cenomanian/Turonian paleogeography from (Sewall et al., 2007) is masked over continental areas and remapped onto the curvilinear PALEORCA grid. Given that the bathymetry of the Pacific and Indian oceans is mostly flat (without ridges) and

deep (∼5000 m) in Sewall et al. (2007), we have decided to merge the more recent bathymetric dataset of Muller et al. (2008) with the land/sea mask and continental margins of Sewall et al. (2007) to create a more realistic deep bathymetry. Handmade corrections are then applied to the bathymetry to ensure that narrow straits (e.g. West Africa) are at least two grid points wide. This avoids the implementation of specific hard-coded schemes in the model to represent water advection across those straits.

**Geothermal heating**. The model modern boundary conditions include a realistic map of geothermal heating, whose impact on

the ocean abyssal dynamics is non-negligible (Emile-Geay and Madec, 2009). As the geothermal heat flow $q(t)$ can be related to the ocean lithosphere age $t$, we construct a geothermal heating map for the Turonian based on the reconstruction of the Turonian ocean crustal age of Müller et al. (2008). We derive the geothermal heatflux using the age-heatflux relationship of the (Stein and Stein, 1992) model.



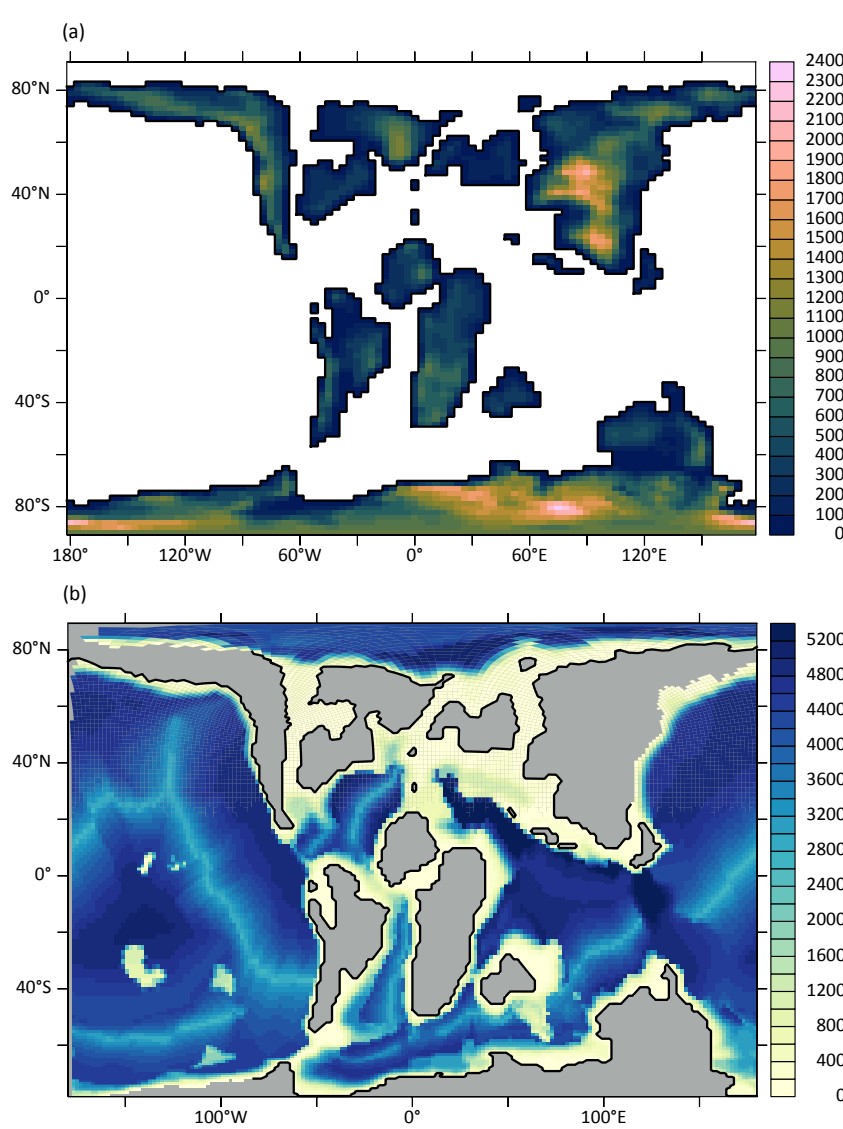

**Figure 19.** Topography (a) and bathymetry (b) (in meters) imposed for the Cretaceous simulation.





For $t \leq 55$ Ma:

$$q(t) = 510t^{1/2} \tag{2}$$

For $t > 55$ Ma:

$$q(t) = 48 + 96e^{-0.0278t} \tag{3}$$

In regions where crust age is missing in (Muller et al., 2008) dataset, we impose the minimal geothermal heat flux obtained with the above equations. The geothermal heating map created is then remapped on the PALEORCA grid and we limit the maximal heat flux to 400 mW.m$^{-2}$ because the Stein and Stein (1992) parameterization becomes singular for young ages (Emile-Geay and Madec, 2009).

**Tides**.Modern boundary conditions also include forcings of the dissipation associated with internal wave energy for the M2 and K1 tides component - see (de Lavergne et al., 2019). Here, in the absence of an appropriate model to estimate M2 and K1 dissipation values for the Cretaceous, we neglect this forcing and set these fields to zero.

**Initial conditions**.Salinity is initialized as a uniform value of 34.7 per mil equal to the modern mean ocean value (Lunt et al., 2017). The ocean initial temperature follows a slightly altered formulation of the zonally symmetric temperature distribution proposed by Lunt et al. (2017):

if $depth \leq 1000$ m :

$$T = 10 + \frac{(1000 - depth)}{1000} 25\cos\varphi \tag{4}$$

if $depth > 1000$ m :

$$T = 10$$

These initialization temperatures are quite high, as the ice-free Cretaceous periods have been shown to have far much warmer oceans than present, associated with high atmospheric pCO2. No sea ice is prescribed at the beginning of the simulation.

## 5.3 Continental boundary conditions

**Topography** We use the Cenomanian/Turonian topography from Sewall et al. (2007) but modify the Antarctic continent because the PALEORCA mesh is not defined beyond 78°S, which corresponds to the maximal extent of the modern Antarctic coastline (including ice shelves). This requires that the Antarctic coastline reaches at least the same latitude for paleoclimate simulations, which is not the case in the Cenomanian/Turonian continental configuration of Sewall et al. (2007). We therefore impose the maximal Antarctic topography proposed by **?** for the Eocene-Oligocene, which has the advantages of meeting the required Antarctic coastline criterion and is a relatively recent estimate of a deep-time Antarctic topography, which remains generally poorly constrained. The final topography/bathymetry is shown on Fig. (19b). LMDZ requires a 10-minute global elevation dataset to compute subgrid-scale parameters (elevation standard deviation, slope, peaks and valleys) that are needed



| Latitudinal range (°) | Plant Functional Type |
|---|---|
| 0 – 15 | 75% Tropical broadleaved evergreen |
| | 25% Tropical broadleaved summergreen |
| 15 – 20 | 100% C3 grass |
| 20 – 35 | 100% Bare soil |
| 35 – 50 | 70% Temperate broadleaved evergreen |
| | 30% Temperate needleleaved summergreen |
| 50 – 80 | 100% Boreal broadleaved summergreen |
| 80 – 90 | 100% Boreal needleleaved summergreen |

**Table 4.** Prescribed Plant Functional Types as a function of the latitude.

to compute surface roughness length as well as to activate orographic drag parameterizations (Lott and Miller, 1997). As the paleotopography provided by Sewall et al. (2007) is 2.8°x1.4° in resolution, we simply artificially upscaled it to a 10-minute resolution through the use of cdo remapping tools (Schulzweida, 2019).

**River routing** The river routing system (i.e. length of rivers, direction of flows, locations of river outflows) is constructed by first upscaling the low-resolution topography of Sewall et al. (2007) to a 1°x1° resolution and second using the (Wang and Liu, 2006) algorithm within the QGIS software (QGIS Geographic Information System. Open Source Geospatial Foundation

Project. ). This algorithm ensures that inland depressions are filled and therefore that all continental freshwater is routed to the ocean in the coupled model.

**Vegetation** In the absence of globally consistent vegetation reconstructions for the Turonian, the imposed vegetation follows a very simple latitudinal distribution of Plant Functional Types (4).

### 5.4 Results of the case-study experiment

#### 5.4.1 Equilibrium

Time series of SST and T2M shows the strong model departure from the initial conditions in the first centuries of the simulation. After 3,000 years, SST and T2M are adjusted at 28.3°C and 26°C, respectively (Fig. 20e). OHC of the first 300 meters is stabilized as well. The long-term adjustment of the ocean is, as for our initial pre-industrial experiment, driven by the initial radiative unbalance resulting from the initial ocean temperatures and the greenhouse forcing. The vertical cross sections at

initial and final states of the experiment shows the spatial patterns of adjustment of ocean temperature and salinity (Fig. 20abcd). The constant salinity together with the latitudinal profile of temperature imposed as initial conditions also induce a latitudinal gradient in seawater density, and thereby an adjustment of ocean dynamics. Surface salinity and temperature are also adjusting driven by the precipitation minus evaporation balance (PE) and freshwater inputs from river routing (see next section) + radiative balance. Contrary to surface signals, the total OHC shows that the deep ocean is still adjusting after 3,000

years (Fig. 20e), showing the need for very long integrations when one model deeptime paleoclimates.





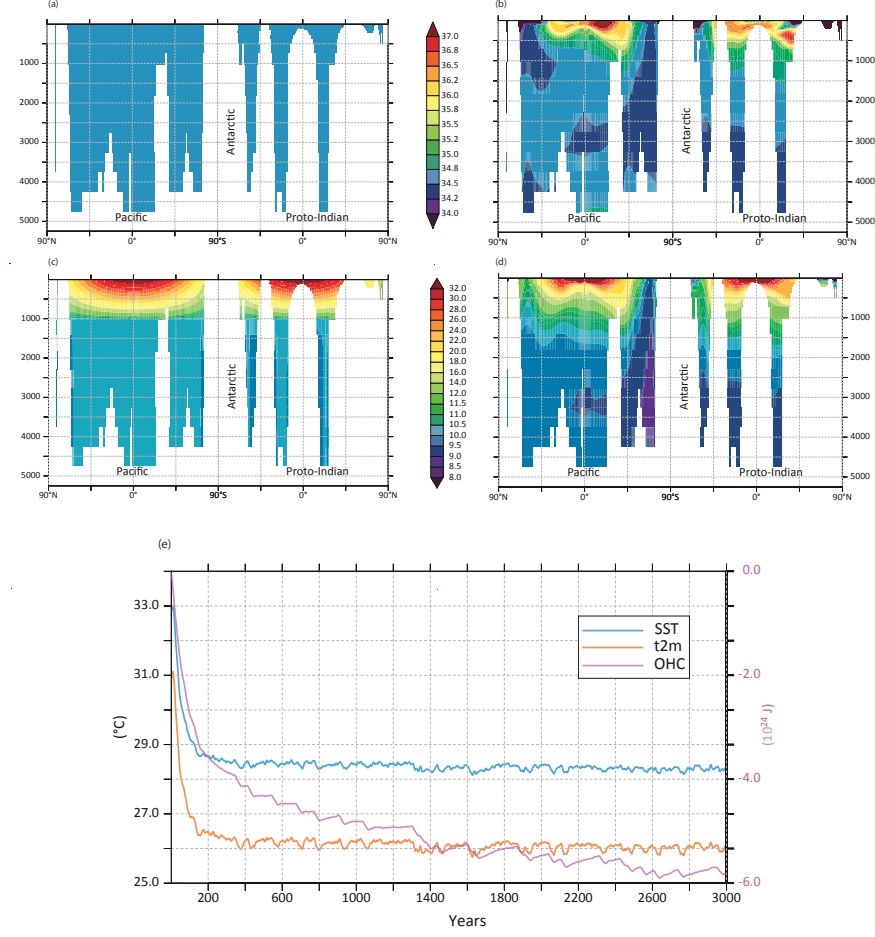

**Figure 20.** Adjustment of the Cretaceous simulation over 3,000 years. Latitudinal cross-section of initial (a,c) and final (b,d) states of salinity (top, PSU) and temperature (bottom, °C) for the Cretaceous simulation. e. Trend in SST, t2m and total OHC.

### 5.4.2 Mean surface climate

The elevated values of SST and T2M at equilibrium are primarily explained by the high atmospheric $CO_2$ concentration prescribed. In the ocean, warmest temperatures (> 36°C) are found in the equatorial band in the Atlantic and Indian oceans as well as in the western Pacific (Fig. 21ab), whereas the eastern equatorial Pacific is cooler by a few degrees. The Southern Ocean is relatively warm (> 10°C) as is the North Pacific while the Arctic Ocean exhibits the coldest surface temperatures (reaching less than 8°C at the highest latitudes). There is no sea ice formed in the polar oceans. In continental areas, the radiative impacts of the high atmospheric $CO_2$ concentration and the absence of ice sheets also substantially affect the mean annual t2m. Values > 40 °C are found in equatorial Africa, South America and southeastern Asia, whereas below 0 °C mean annual continental t2m are only located in Antarctica (Fig. 21b). The winter season yields well below freezing t2m over large parts of the NH






high latitudes, with lowest values in northern Siberia. Similarly, the whole Antarctic continent suffers below freezing t2m, with values below -16°C over large parts of the continental interior). In the summer however, Antarctic t2m are mostly above 20 °C and reach up to more than 28 °C close to the modern Weddell Sea (not shown). Tropical continents are affected by extremely high summer t2m reaching more then 44°C whereas NH high latitudes exhibit the lowest summer values with < 16°C in northern Siberia.

The mean annual precipitation-rainfall (PE) shows expected patterns across the globe (Fig. 21d), including the double ITCZ as in the preindustrial and historical simulations. Net evaporation areas located at subtropical latitudes under the descending branch of the Hadley circulation and net precipitation areas located in the equatorial band (except in the equatorial Pacific and Atlantic) and in the mid- to high-latitudes. Note the particularly dry tropical continental regions of Africa and South America, which are the results of their more southern position during the Cretaceous (the same is true for South Asia in the Northern

Hemisphere).

PE generally explains well the patterns of open ocean sea surface salinities (SSS, Fig. 21c). Highest SSS are thus found in the subtropical areas whereas lower values are found in the mid- to high-latitudes and in the equatorial band. This spatial SSS distribution is altered by the supply of water from continental rivers. The very high amount of precipitation occurring in the equatorial band over India and Northern Africa indeed increase the freshening of nearby coastal waters. Low salinity

waters are also found in the Australo-Antarctic sector of the Indian Ocean and in the Arctic Ocean, where large paleorivers can supply significant amounts of freshwater to the ocean. The relative isolation of the Arctic from the global ocean limits possible exchanges of water and therefore may amplify the freshening of this basin. The Pacific sector of the Southern Ocean is more saline than other corresponding high latitudes oceanic regions (North Pacific, Atlantic and Indian parts of the Southern Ocean) with SST of similar or lower values. The winter cooling of the dense surface waters in this region therefore leads to

deep water formation as shown by the winter deepening of the mixed-layer depth (Supp. Fig. XX). Contrary to the modern, there is no deep water formation in the Northern Hemisphere, in relation with the peculiar Turonien paleogeography, in which the northern Atlantic is only partially open and NH landmasses reach low latitudes.

This simulation, and by extant the overall response of IPSL-CM5A2 to a Cretaceous paleogeography without ice sheets and elevated CO2 levels, would need to be evaluated in more details against previous work and proxy data. Still, the basic analysis

presented here demonstrates that the mean climate simulated by our model shares many similarities with that simulated by other investigations of Cretaceous warmth with coupled models (Niezgodzki et al., 2017; Tabor et al., 2016; Upchurch et al., 2015). It therefore suggests that our goal of adapting the IPSL ESM to deep-time boundary conditions has been reached with success.



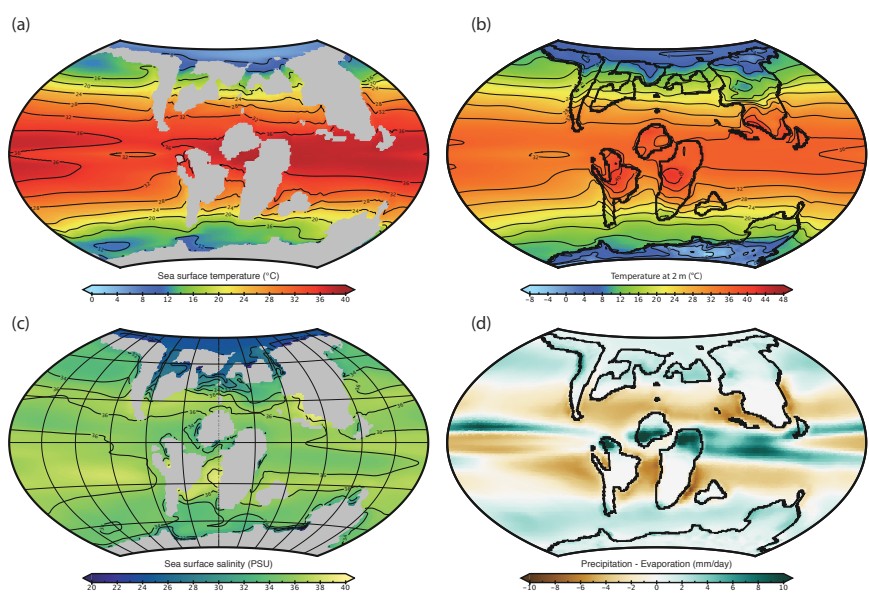

**Figure 21.** Climate mean state of the Cretaceous experiment after 3000 years. a. SST (°C), b. t2m (°C), c. SSS (PSU), d. PE (mm day$^{-1}$).



## 6   Conclusions

This article aimed at providing a detailed description of the building of the IPSL-CM5A2 ESM, developed with the aim of performing multi-millennial earth system simulations. We expect IPSL-CM5A2 to be used mostly for deep-time paleoclimate modelling and, given its reasonable computing performances, attempts of transient simulations for the Quaternary and the future. Starting from IPSL-CM5A, our goal to obtain a faster model with reduced biases has been reached successfully. IPSL-CM5A2 pre-industrial integration shows a reduced cold bias, no drift in the OHC after 1500 years, and reasonable energy leaks

and salinity drift. Despite persistent biases in the tropics, the historical runs also show a more vigorous AMOC, together with better-located westerlies. The simulation of global NPP, although still underestimated, has been largely improved. Our case-study for the Cretaceous shows the numerous hurdles, often undocumented, that need to be crossed for setting-up a climate simulation for deep time.

Documenting the strength and weaknesses of IPSL-CM5A2 stimulates questions for the future of paleoclimate modelling.

Although numerous interdisciplinary studies can be triggered by efficient climate modelling at the frontier of geology, climatology and biology (see e.g. Haywood et al. (2019) for a synthesis), the question of the stability of biases with radically different boundary conditions remains open. Recently, Krinner and Flanner (2018) opened a path by exploring the stationarity of ESMs large scale biases when forced by four-time atmospheric $CO_2$ concentration. Answering the same question for paleoclimate contexts in which the boundary conditions, such as land-sea mask, topography and bathymetry, insolation and greenhouse gas

forcing, are so different than present that they impose very strong forcings on the climate system, remains to be done. Until this challenge is tackled, the use of a computing-efficient ESM to run sensitivity experiments to individual forcing factors, for which the geologic record provides good constraints, appears to be the key.

On the technical point of view, the ongoing race to higher resolution in ESMs asks for consistency between code architecture and supercomputer design. For example, a simple comparison of computing performances of IPSL-CM5A2 and IPSL-CM6-

LR, the higher resolution IPSL model designed for CMIP6, shows that the latter requires ∼8 times more computing resource than IPSL-CM5A2 to run a simulation of a given length (Fig. A10). Specifically, for a 1000-year experiment, IPSL-CM6 require more than 1.6 millions of computing hours, simulating 16 SYPD on more than 1000 cores. This model is being used for paleoclimate snapshots experiments, like in the PMIP project, but its use for long, multi-millennial simulations that are typical of deep-time paleoclimate studies is compromised by these performances and requirements. Similar constraints have

led colleagues from other climate modelling group to keep updating and using climate models at the resolution they were originally designed more than 15 years ago (Valdes et al., 2017). Given the apparent contradiction between (i) the need for long integrations in deep-time climate modelling, (ii) the enormous resource requirements of modern ESMs, and (iii) the limited number of computing time available on supercomputers, this strategy may well be the most appropriate, in the absence of a technological leap. At IPSL, the next generation of ESMs will likely include an atmospheric model with a new dynamical core

(LMD-DYNAMICO) for which scaling will be far better than LMDZ. The very recent version of NEMO (version 4) might also help breaking this computing lock, as the code now benefits from high level parallelization. Still, there is a supplemental challenge for paleoclimate modelling: numerous developments have been made in the recent years in the different components





of IPSL ESMs, with the inclusion of passive tracers in the ocean (Arsouze et al., 2007), the water isotopes in the atmosphere (Risi et al., 2010), and the explicit resolution of chemical photoreactions in the stratosphere (Jourdain et al., 2008). All of them
appear to be particularly relevant for paleoclimate studies (Botsyun et al., 2019; Sepulchre et al., 2014; Szopa et al., 2019), but most drastically reduce the computing efficiency of the model. A trade-off between the number of tracers and/or processes to be included in the model, its spatial resolution, and the length of simulations required for paleoclimate simulations will thus have to be found in the next years to setup relevant experiments designs in the deep-time climate modelling field.

*Code availability.* LMDZ, XIOS, NEMO and ORCHIDEE are released under the terms of the CeCILL license. OASIS-MCT is released
under the terms of the Lesser GNU General Public License (LGPL). IPSL-CM5A2 code is publicly available through svn, with the following command lines: svn co http://forge.ipsl.jussieu.fr/igcmg/svn/modipsl/branches/publications/IPSLCM5A2.1_11192019 modipsl
cd modipsl/util;./model IPSLCM5A2.1
The mod.def file provides information regarding the different revisions used,namely :

- NEMOGCM branch nemo_v3_6_STABLE revision 6665
- XIOS2 branchs/xios-2.5 revision 1763
- IOIPSL/src svn tags/v2_2_2
- LMDZ5 branches/IPSLCM5A2.1 rev 3591
- branches/publications/ORCHIDEE_IPSLCM5A2.1.r5307 rev 6336
- OASIS3-MCT 2.0_branch (rev 4775 IPSL server)

The login/password combination requested at first use to download the ORCHIDEE component is anonymous/anonymous. We recommend to refer to the project website: http://forge.ipsl.jussieu.fr/igcmg_doc/wiki/Doc/Config/IPSLCM5A2 for a proper installation and compilation of the environment.

*Data availability.* All model outputs analysed in this paper are available as netcdf files on the zenodo repository with the DOI 10.5281/zenodo.3549769. We also published a jupyter Notebook gathering most of the scripts used to create the figures. This notebook is embed-
ded in a binder container that allows anyone to run it and analyse the code. It is published on the zenodo repository with the DOI : 10.5281/zenodo.3549652. Most of the figures were done using NOAA pyferret within jupyter notebooks, thanks to the ferretmagic add-on developed at LSCE by Patrick Brockmann. Ferret is a product of NOAA's Pacific Marine Environmental Laboratory. (Information is available at http://ferret.pmel.noaa.gov/Ferret, distributed under the Open Source Definition. The Jupyter Notebook is an open-source web application. Most colored figures in this paper were made with perceptually uniform, colour-vision-deficiency friendly sci-
entific colormaps, developed and distributed by Fabio Crameri (Crameri, 2018a, b), to prevent visual distortion of the data. We also used the C-ESM-EP (CliMAF Earth System Model Evaluation Platform) developed at LSCE by Jérôme Servonnat https://github.com/jservonnat/C-ESM-EP/ to evaluate the model: https://vesg.ipsl.upmc.fr/thredds/fileServer/IPSLFS/pselpuch/C-ESM-EP/CM5A2vsDATA_pselpuch/C-ESM-EP_CM5A2vsDATA.html. C-ESM-EP is based on CliMAF (Climate Model Assessment Framework) an open-source library distributed with a GPL-compatible license. We acknowledge the NCAR Climate Analysis Section's Climate Variability Diagnostics





Package (Phillips et al., 2014) for some figures http://www2.cesm.ucar.edu/working-groups/cvcwg/cvdp.Detailed El-Niño diagnoses can be found at : https://skyros.locean-ipsl.upmc.fr/~ggalod/IPSLCM5A2/.

## Appendix A

### A1 Setting up IPSL-CM5A2 configuration

Before making our choice regarding the new model configuration, we explored the computing performances of two configu-

rations of the IPSL coupled model, differing by the spatial resolutions of the atmospheric and oceanic components. The first configuration keeps the spatial resolution of IPSL-CM5A, i.e. 3.75 °x1.875 ° and 39 vertical levels in the atmosphere (hereafter 9696L39), and around 2° in the ocean (with a refinement to 0.5 ° in the tropics, hereafter ORCA2). The second configuration has higher spatial resolution: the atmosphere is at 1.25 °x2.5 ° and 79 vertical levels (hereafter LMDZ-144143L79) and has a 1 ° resolution in the ocean (NEMO-ORCA1). This last version is the one used for the CMIP6 project. Here we analyze the com-

puting performances of the separate components to optimally allocate computing resources and determine the best trade-off between resolution and computation time. Scalability information for each component is obtained from LUCIA (Maisonnave and Caubel, 2014) which is directly incorporated in OASIS3-MCT. LUCIA measures at each time step the time spent for (i) component calculations, (ii) variables received/sent from/to other components, and (iii) interpolations. This method has the advantage of measuring the exact scalability of the components during the coupled system production phase. The maximum

speed of the coupled system is approximately the speed of the slowest component because the model is run on concurrent mode (i.e. both components run at the same time because they require fields from the previous coupling time step). Fig. A2 shows that despite tests with more than 500 cores, LMDZ and NEMO scalability with configuration 2 remains low, with maximal performances reaching 8 simulated years per day (SYPD) for LMDZ144143L79 and 12 SYPD for NEMO-ORCA1. Scalability is much higher in configuration 1, with NEMO saturating at more than 100 SYPD with 200 cores and LMDZ reaching 56

SYPD with 256 cores (32 MPI processes x 8 OMP threads). These results lead us to use the first configuration as a basis for designing a fast version of the IPSL coupled model.



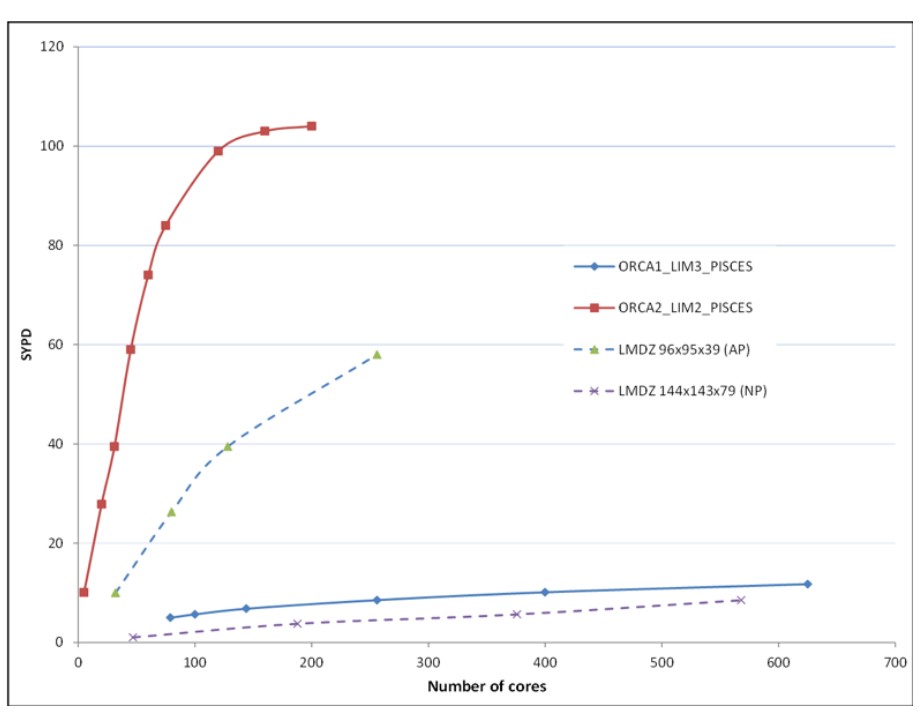

**Figure A1.** Computing performances for LMDZ and NEMO component at two resolutions in the early tests made on supercomputer CURIE.





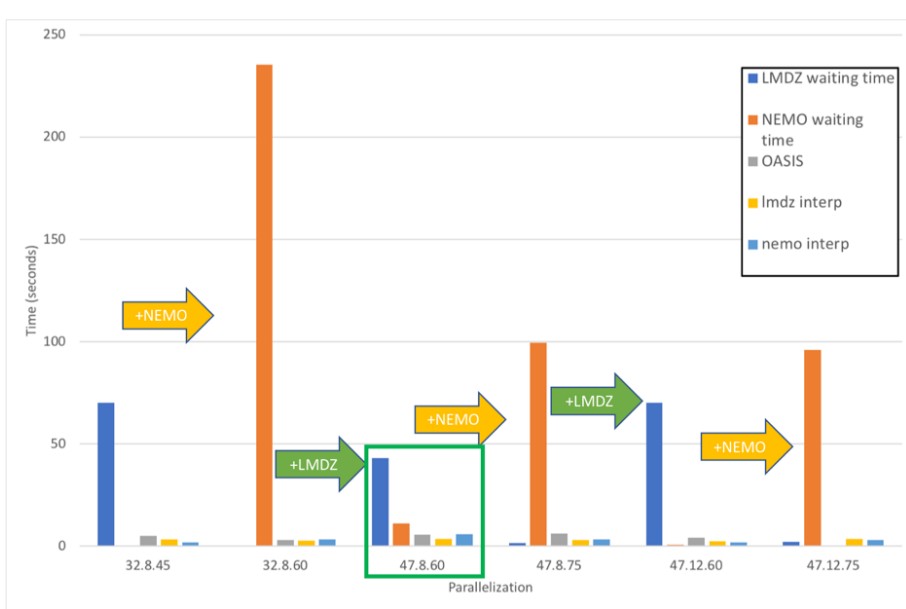

**Figure A2.** Waiting time of the model components and computation time of the coupler and interpolations, as a function of incremental resources provided to each component. The XX.X.X code on the x-axis indicates successively the number of MPI processes, OMP threads for LMDZ and MPI processes for NEMO. This shows that most efficient configuration is the one with 47 MPI and 8 OMP dedicated to LMDZ and 60 MPI dedicated to NEMO.





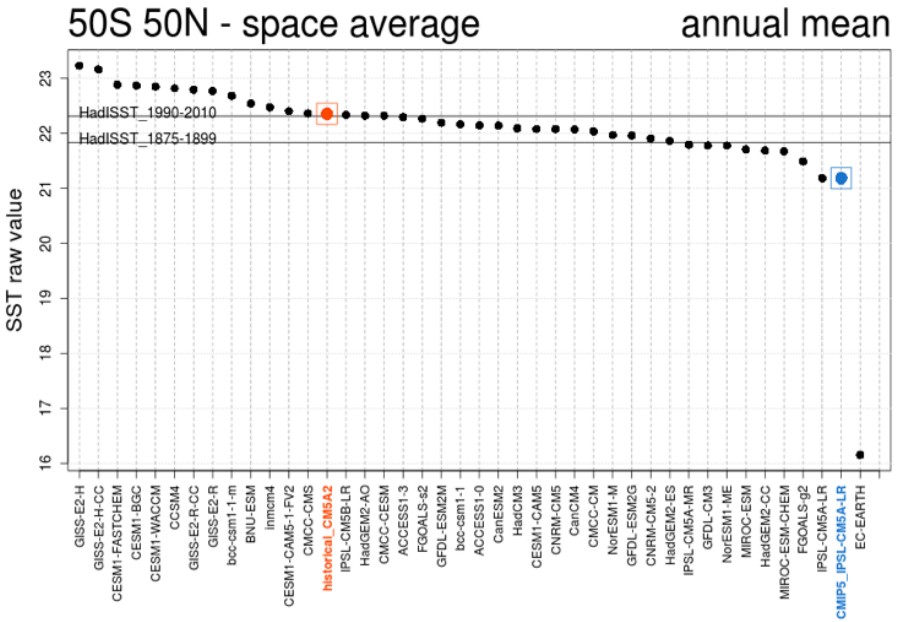

**Figure A3.** Simulated SST of the historical experiment compared to results from CMIP5 models

## A1    IPSL-CM5A2 SST compared to CMIP5 models.



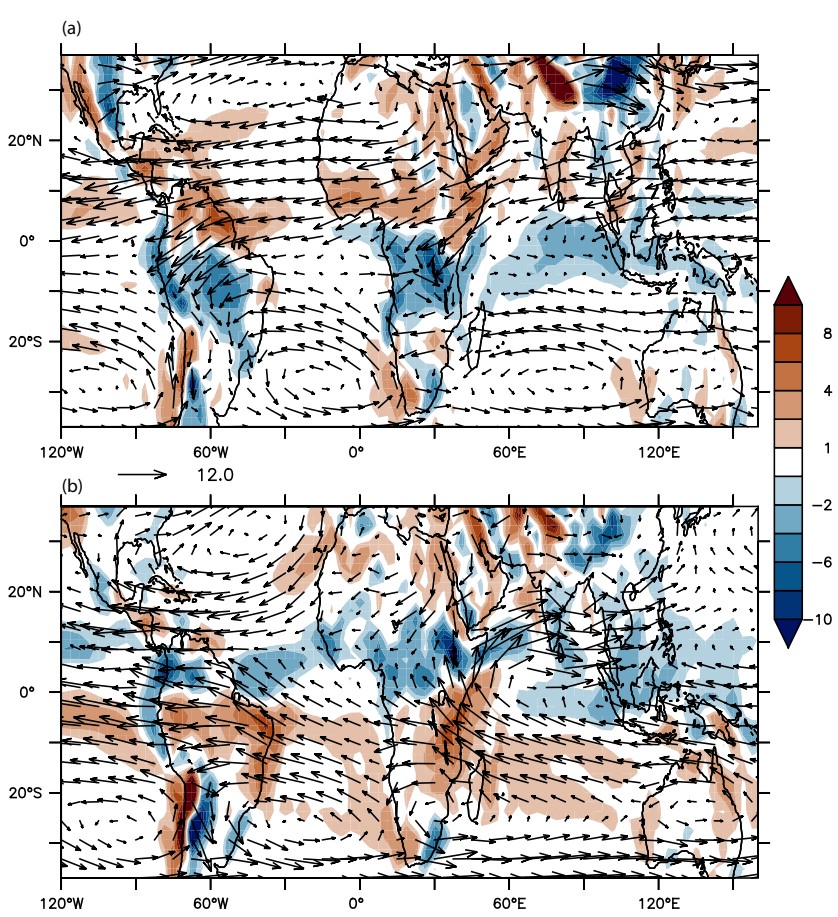

**Figure A4.** 850 hPa wind divergence for DJF (a) and JJA (b).

## A2    Wind divergence at 850 hPa.



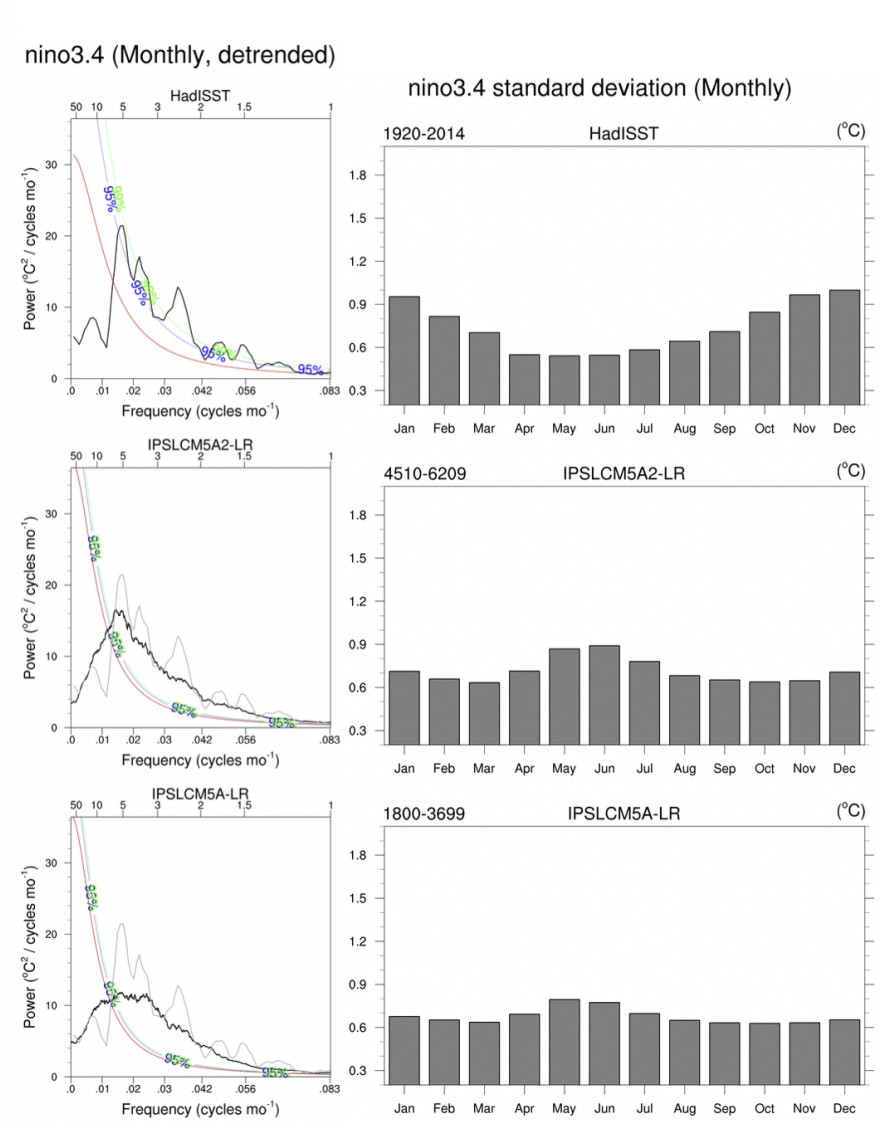

**Figure A5.** (Left) Power spectra of the Niño3.4 index (SST in 5°N-5°S, 170°W-120°W) in variance-conserving form, in observations (top, HadISST, 1920-2016), and in the PREIND simulations of (center) IPSL-CM5A and (bottom) IPSL-CM5A2. The best-fit first-order Markov red noise spectrum (red curve) and its 95% (blue curve) and 99% (green curve) confidence bounds are shown on each panel. (Right) Monthly standard deviation of the Niño3.4 index for the same observations and simulations.

## A3   El Niño analysis.



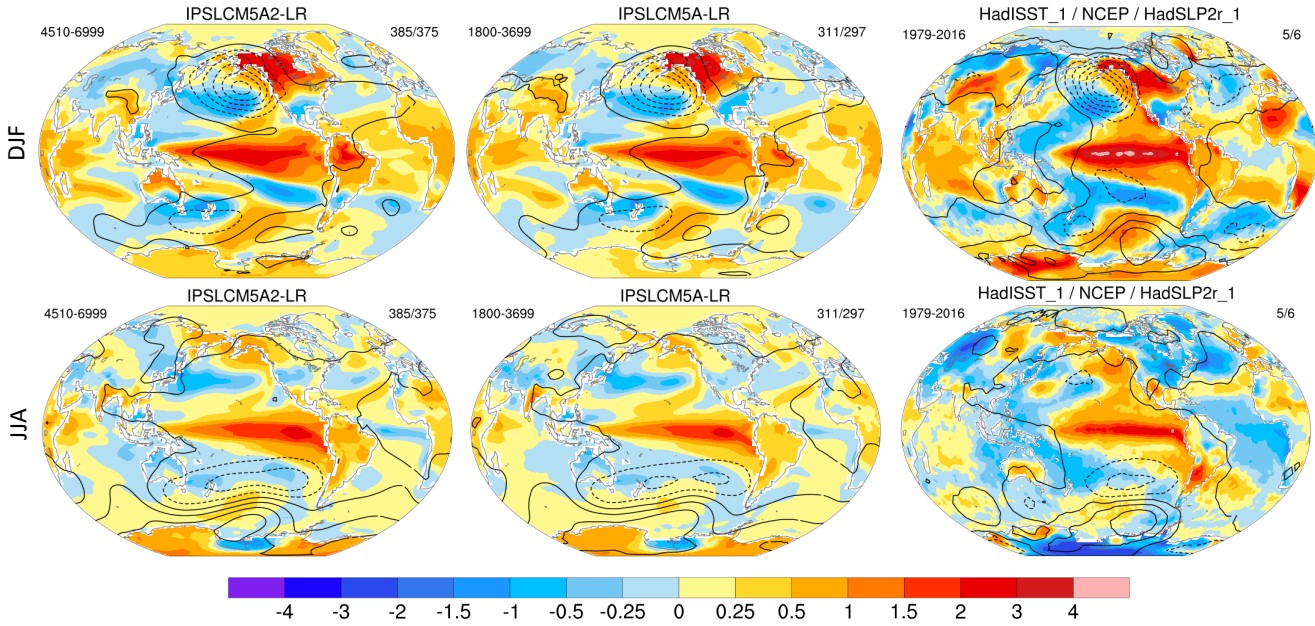

**Figure A6.** Composite of all years greater than 1 standard deviation (El Niño) and all years less that -1 standard deviation (La Niña), based on the December Niño3.4 index (SST in 5°N-5°S, 170°W-120°W) smoothed with a 3-point binomial filter; for observations (1979-2016 period), and for the PREIND simulations of (left) IPSL-CM5A and (center) IPSL-CM5A2. The number of El Niño/La Niña events composited is shown in the right subtitle. Temperatures are color shaded and in units of Celsius and are show the SST in the ocean and the 2m temperature over the continent. Sea level pressure is contoured from -16 to 16hPa by 2hPa; negative contours are dashed. Top row is for DJF, almost simultaneous with the December Niño3.4, while bottom row is for JJA preceding December Niño3.4 by about 5 months.

**A4   Cretaceous seasonal temperatures and mixed-layer depth**




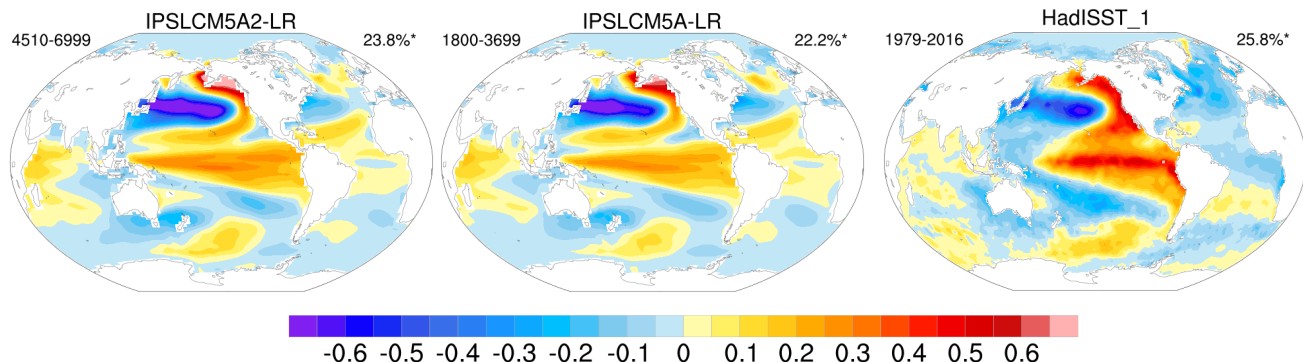

**Figure A7.** SST patterns for the Pacific Decadal Oscillation (PDO). The PDO is defined using monthly leading principal component of North Pacific (20:70°N, 110°E:100°W) area-weighted SST* anomalies, where SST* denotes that the global mean SST anomaly has been removed at each timestep. Pattern created by regressing SST anomalies (in °C) onto the normalized PC timeseries.

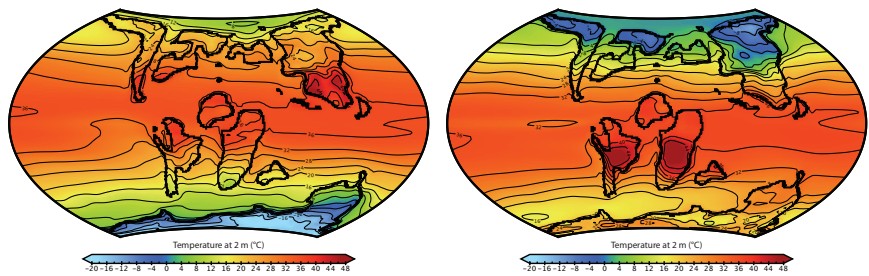

**Figure A8.** Simulated JAS and JFM near-surface temperature after 3000 years of run for the Cretaceous experiment.

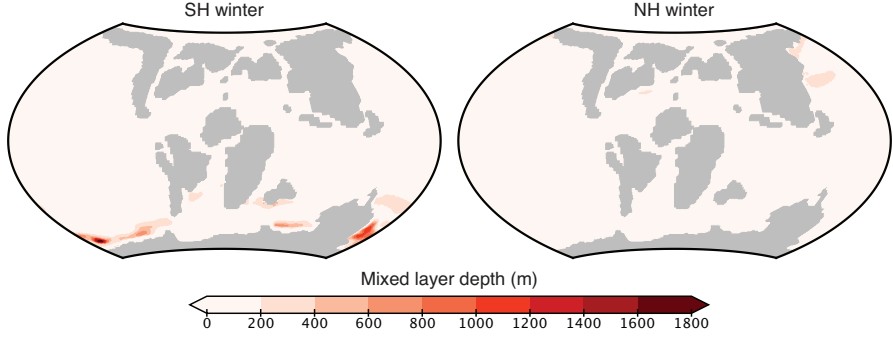

**Figure A9.** Simulated JAS and JFM mixed-layer depth after 3000 years of run for the Cretaceous experiment.

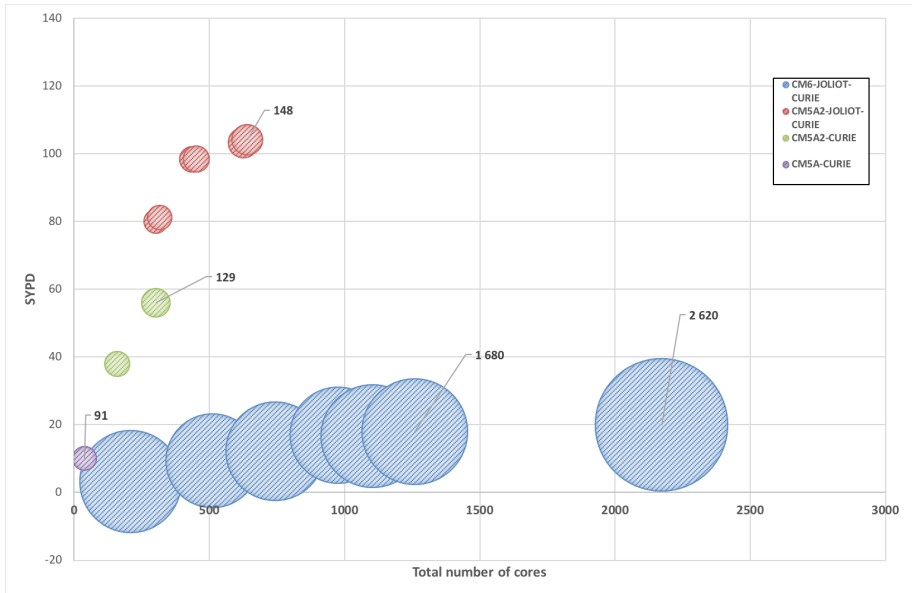

**Figure A10.** Scalability, i.e. SYPD as a function of parallelization, of IPSL-CM5A2 on CURIE and JOLIOT-CURIE supercomputers compared to IPSL-CM6-LR (blue disks) on JOLIOT-CURIE. The surface of the disks is proportional to the computing cost (CHSY).

## A5    IPSL-CM5A2 computing performances compared to IPSL-CM6-LR

*Author contributions.*  PS and AC designed and ran the pre-industrial and historical numerical experiments. JBL designed the Cretaceous boundary conditions and ran the Cretaceous experiment. JBL, YD, FF and PS analysed the Cretaceous experiment. FH and PS designed model tuning. GG, DS, JM, VE-P carried on ocean mean-state and variability analyses. GG carried out the CVDP diagnoses. LB and JBL analysed ocean productivity. OB, GG, FH and PS discussed the energy conservation of the model. AC, CE, JG, AC, YM helped setting-up the model configurations and analyzed computing performance. FF and DT analysed monsoon signals of the model. A-C S analysed the response over Indonesia. JS run the C-ESM-EP diagnoses. AC, CE, YM, JG described the model components. All co-authors contributed to the writing of the manuscript.

*Competing interests.*  The authors declare no competing interests.

*Disclaimer.*  The simulations analyzed in this paper required approximately 500,000 CPU hours, emitting approximately 1.7 teqCO2.



*Acknowledgements.* Authors wish to acknowledge Jean-Louis Dufresne for fruitful conversations at the early stages of the model setup. The setup of the model benefited from the pioneer work of Gurvan Madec and Alistair Sellar on the ocean grid. This work was granted access to the HPC resources of TGCC under the allocation 2019-A0050102212 made by GENCI. J.-B. L. and Y.D. acknowledge the support of the project Anox-Sea funded by the Agence Nationale de la Recherche (ANR) under the grant ANR-12-BS06-0011-03. S.S. and P.S. ackowledge the support of the PALEOx project, funded by ANR under the grant ANR-16-CE31-0010.






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
