# Peer review of "IPSL-CM5A2. An Earth System Model designed for multi-millennial climate simulations"

_Geoscientific Model Development, 2019_

## Referee Comment (RC1) · Anonymous Referee #1 · 3 Jan 2020

Review

IPSL-CM5A2. An Earth System Model designed for multi-millennial climate simulations

By Pierre Sepulchre et al.

Summary:

This paper provides a comprehensive description and analysis of the IPSL-CM5A2 model developed primarily for deep time paleoclimate applications, and/or multi-millennium simulations where very long integrations are required. It includes 1) technical and performance descriptions, including model tuning decisions, 2) documentation by comparison to observations, and 3) a deep time case study designed to show this model is fit for purpose.

[Figure]

Overall Comments:

I applaud the authors for this very impressive and detailed work. The paper covers a lot of ground and will appeal to a diverse audience including computational scientists and software engineers (technical and performance analysis), model developers (tuning decisions and model details), climate scientists (model validation against observations and similar earth system models), and finally, paleoclimate modellers and developers (deep time case study).

I do have a number of suggestions, however, which are detailed below. Primarily, they are mostly minor, but important to improve the readability of this manuscript. Inconsistencies in organization and figures must be addressed before full publication.

As the focus in main paper on scientific improvements, the authors may want to consider moving the performance-related discussion to the appendix, and climate state figures back into the main text. Another idea might be to keep the appendix material themed; something like (a) performance analysis and (b) modes of variability. Although both of these topics are very important, if the main paper covers the main model improvements and climate state analysis, it may help to tighten up the manuscript.

Lastly, although the authors note that the case study is not meant to be an evaluation of the IPSL model with the Cretaceous time period, I would like to see more discussion on how it compares to proxy data (or other published work).

Specific Comments:

Introduction:

Somewhere in the introduction, the acronym IPSL should be defined.

Lines 20-66: The CESM system has been run for multi-millennia for CCSM3, and now, currently with CESM1 under the TraCE project. Although this is not continuous in that the model is stopped, adjusted, and restarted for separate timeslices, it is a multi-millennia simulation nonetheless. For CCSM3:

http://www.cesm.ucar.edu/working_groups/Paleo/documentation/sims-projects/trace-21ka/ and Feng He Phd dissertation (U. Wisconsin, 2011); for CESM1, contact Dr. He.

Line 39: Define "branched". Even for climate modellers, this term can be a source of confusion.

Paragraph Line 67: For those not familiar with the IPSL model, a better overview of the original model is needed for the introductory context, i.e. a mention of all model components (are there not sea ice and river runoff in NEMO?). The Dufresne et al reference is not enough.

From IPSL-CM5A to IPSL-CM5A2:

Section 2.1. It would be very useful to see a table summarizing this information including how each component has changed (or not changed), including resolutions.

Table 1: What is the significance of the bolded information? Please note in the table caption. Also, JOILOT-CURIE has not been mentioned yet. Adding the heading "supercomputer" to this column would help.

Sections 2.2 and 2.3. These sections would work just as well as an appendix.

Tuning

Table 2 needs units. How are the Trenberth estimates used here?

Lines 268 and 280: change "insure" to "ensure".

Comparison with observations

Line 323. There is no figure showing spatial biases, so please add "not shown".

Line 351. "every region..."

Line 377, Figures 7, 8, 9. It isn't clear when you are referring to Figure 7 or 8, or Figure 7 or 9 in the text, thus making these sections hard to follow. Also, Line 377 refers to Fig

15, when I think the authors mean Fig 9. Please review text and figures for consistency. Also, please make IPSL-CMA titles in the figures consistent with one another. Figs 8/9 uses "LR" when others do not.

Figure 10. This is more than just a Hovmueller (only panels c and d).

Paragraph Line 396 and Figure 11. Consistently refer to figures here as in the manuscript, i.e. use the lettering a,b,c,d. Also, please use the dataset name in the figure caption, rather than simply "data". For surface temperature, I assume this is HadiSST? What is used for observation for the salinity, WOA2013? I don't see this information in text either.

Line 426- 429 and Figure 15a. The figure caption needs to be improved. It is clear that 15b, the colors are the differences from the two models, but not so with 15a. If the purpose is to highlight the improvement in AMOC for 15a, then a figure showing the timeseries of AMOC, maximum below 500, between the two models, could be an alternative figure.

Section 4.2.4 and Figure 17. Consistency between this figure and others in this manuscript regarding lettering or positional references, and labeling of models is needed. Also, the caption needs to be improved to include years and model names. For the text, some discussion with observations in needed for context.

Figure A5. What is the difference between the gray and black lines in the power spectra? Also, labelling of appendix figures needs to be consistent with the rest of the manuscript.

Figure A6. Consistency in labelling.

Deep time case study

Line 503. "..ran a 3000-year long..."

Section 5.1 and Figure 18. I am not sure what I am looking at in Figure 18a. Are

the white areas supposed to be the grey-shading for the refinement? (This could be a document format issue so, perhaps noting the refinement areas with a black box rather than colored shading would be better). Also, for Figure 18b, my recommendation would be to make this with the cretaceous continental outlines, not modern. I think the discussion of pole and refinement placement would be much clearer to readers who have never modified grids for deep time if the paleo outlines were used.

Section 5.2. What did you use for other important forcings, such as solar input and methane? Should we assume that all else is based on pre-industrial forcings? Does this model prescribe or predict aerosol emissions? Dust, sea salt, carbon from fire, and other biogenic aerosols such as secondary organic aerosols (isoprene, etc), may also be extremely important climate forcings, even for deep time.

Line 580. I suspect the "?" was meant to stand in for something? Please fix.

Line 582 and Figure 19. Should this be Fig 19a? Given that 19b is discussed first, a and b should be swapped in Figure 19.

Line 592. Have the authors worked on applying the dynamic vegetation to the Cretaceous? Dynamic vegetation capabilities are highly valued in the deep time modelling community.

Line 607 and Figure 20. Define T2M. Is it 2-meter air temperature, same as "t2m"? Please make consistent throughout the manuscript. Also, I would not consider 20e a "trend" plot as the figure caption states. Although trend can certainly be implied, but the variable being plotted is a whole value as it varies through time, the unit being degC (or Joule).

Figure 21. Assuming this is annual mean. How many years (the final 100?) went into this plot?

Line 635. Fill in Supplemental Figure XX. Is this referring to figures A8 and A9?

Figure A8 and A9. Please label consistently. Briefly note/explain why you chose JMA

and JAS and northern/southern hemisphere winter.

Conclusions

Line 645. "This article aims..."

I would recommend changing the title of this section to "Conclusions and Discussion" because of 1) the inclusion of new appendix material and 2) the relevance of the new material to the overall discussion on challenges for (deep time) paleoclimate modelling.

Appendices

An appendix section describing the many acronyms in this paper would be helpful.

Appendix organization needs to be fixed. There are two A1's. Also, only the first A1 includes text to introduce the appendix figures. A few sentences introducing each appendix section would greatly improve the readability of this appendices.

---

## Referee Comment (RC2) · Dan Lunt (Referee) · 12 Feb 2020

I was looking forward to reviewing this paper because (a) it is exactly the sort of paper that we hoped the existence of GMD would enable and encourage when journal first started, (b) I have been involved in a similar paper describing our own model (Valdes et al, 2017), and (c) it has a section on configuring the model for deep-time simulations, which is of great interest scientifically to me.

I think that this paper will be very useful, providing a definitive benchmark for this model. I hope that the authors start a Special Issue on this model, and submit bug-fixes/developments as they occur, in follow-up papers.

I have some questions/comments (none of which are major), some suggestions for additional work (which I would view as nice but optional), and some minor corrections.

Dan Lunt

**Questions/Comments:**

- Introduction: as well as paleoclimate, there will be interest from the forcings/feedbacks community associated with longrunMIP (e.g. Rugenstein et al, 2019). Would be good to mention this. In general, I think you underplay the interest of this model to the non-paleo community.
- Abstract needs to include the resolution of the model (horizontal and vertical, atmosphere and ocean).
- Line 20: Fully complex (e.g. CMIP6) models don't only carry out snapshots, often transient runs (e.g. RCPs!). More generally, remember the audience for this paper is not just paleo scientists - you should assume an audience that is not familiar with paleoclimates, e.g. avoid terms such as "Quaternary", "last interglacial" without explaining when they were.
- Line 64: I would remove the statistical approach from the transient section, and instead add another two types of approach here, which are:
  (a) "multiple snapshots". Within this you could include e.g. Singarayer and Valdes (2010), and Marzocchi et al (2015).
  (b) statistical approaches based on multiple simulations, e.g. Lord et al (which you already cite), and Simon et al (2017), and Erb et al (2015).
- Line 96: It is not always clear from the description which components are identical to IPSL-CM5A, and which are "updated versions of IPSL-CM5A components"
- Line 116 indicates that "The differences mostly include bug corrections, various improvements in the physics/dynamics interface, and energy conservation. Changes also concern the Emanuel convection scheme, with bug corrections and changes in the upper bound of the deep convection loop.". it would be good to maybe say a little bit more about these, and explain why these changes were made.
- Figure 1 looks pretty, but would include more information if a standard cylindrical projection were used? Or maybe if views towards the Pacific were also added.
- Section 2.1: I like the way that the actual tagged releases of each component are given for the new model and for the old model. This is very good practice. Is there also a version number for the overall model, for both new and old versions? And is there a unique update number for NEMO and OASIS as well (the later section on Code Availability indicates that there is)?
- Line 229: Not sure what "adjusted" means in "provided an adjusted global surface air temperature of 11.3°C"
- Line 230: "Such a cold bias had been depicted in the previous IPSL-CM5A (Dufresne et al., 2013)". What was the magnitude of that bias?
- Line 231: "We targeted to increase global-mean surface temperature by 2.2°C to reach 13.5°C in pre-industrial conditions with IPSL-CM5A2, expecting this value to translate into 15.5°C in simulations with present-day conditions.". This needs a reference to some observational dataset, either to validate the 13.5 preindustrial choice, and/or the 15.5 present-day choice. Also mention the uncertainty in these observations.

- Line 236: "Previous results obtained with LMDZ and IPSL-CM5A show that changing TOA balance by 1 W.m−2 results in a change in temperature of 1 K." I guess you mean "initial TOA inbalance"? Is this the TOA in a PI run, or historical, or….?
- Line 237: "It is also the typical value of the sensitivity of global temperature to greenhouse gas concentration" is rather opaque….do you mean that the overall net feedback parameter, lambda, in this model is ~1.0 W/m2 / K ?
- Line 249: "when two of the bugs mentioned above"…there are several bugs mentioned above…which ones are you referring to?
- Line 257: "respectively 0.029 W.m-2 and 0.023 W.m−2". Is the difference between these two primarily due to interannual variability, or gradual accumulation of energy in the ocean, or bugs in energy conservations, or something else?
- Line 258: What are the causes of the non-conservation of energy?
- In very long simulations, small non-conservation of energy is not a problem, because the energy system is "open" to the outside – i.e. a small non-conservation may have very similar results to slightly increasing the solar constant – the model just finds a new equilibrium. However, the hydrological system is "closed", and so non-conservation can result in substantial drifts in e.g. global mean salinity over very long time periods. How well does the model conserve water, and are any correction factors included to maintain global mean water/salinity content? OK, I see this is addressed later (lines 272-280). Did you consider a salinity correction term? See e.g. the HadCM3 runs in DeepMIP – Lunt et al (submitted to Climate of the Past).
- Line 268: "Such a difference is also small enough compared to the regional biases in sea surface temperature.". How do you define "small enough"?! Would 0.5 degrees by small enough? What about 0.8, or 1.0?! Better to compare with some other models, and just say, e.g. "this is closer to observations than X% of CMIP5 models", or (and) link it to the uncertainty in the observations – 0.3 degrees is probably within error of the observations?
- Supp Info could be used for some more plots, e.g. the old model version of Figure 4(d,e), the old model version of Figure 5(c). 6(a,d) etc etc. I expect you have already made these plots!
- Label all figures, not only in the caption but above the figure panels as well. Some plots are model minus observations, some model minus model, some just model. Clearere labelling would help (I know I could get this info from the caption but this is much slower than labelling the panels!)
- Fugure 15a – presumably this is the new model (not stated in the caption). Would be good to see this also for the old model.
- Line 595: would be good to show evolution of TOA flux, and show a Gregory plot.
- Section 5.4. This section implies that the model worked "first-time", "out-of-the-box" for the new Cretaceous paleogeography. I would be surprised (and impressed!) if this were the case. If not, then it would be good to explain what needed to be done to allow the model to run, e.g. smoothing topography/bathymetry etc.

**Suggestions**

- Line 282: Presumably there is quite a "kick" to the model going from your PREIND forcings to the 1850 forcings; presumably it might have been a good idea to hold the model steady at 1850 CMIP5 forcings for a few decades before starting the 1850-2005 simulation?
- HadCM3 runs at about 100 SYPD these days (see Valdes et al, 2017). it might be nice to compare some metrics with them, in a consistent style e.g. Geckler plots (see Figure 2 in Valdes et al, 2017).
- In addition to the Cretaceous simulation with the new model grid, it would be very informative to re-do a preindustrial control with this new grid, to see how much of a role the modern-specific tweaks play

**Minor corrections**

Line 8: "and" missing.

Line 27: "which" should be "whose"?

Line 150: spell out "phytoplankton" and "zooplankton".

Figure 2 caption: "The surface of the disks" should be "The surface area of the disks".

Figure 4: label panels (d) and (e) as CRF and temp anomalies (just so we don't have to read the caption!).

Figure 19b. Black continental outline is missing for some coastlines.

Line 580. Reference missing.

Figure 21: why suddenly a new map projection?

I was surprised that Jean-Louis Dufresne was not a co-author, given his contributions to the original model?

Erb, M.P., C.S. Jackson, and A.J. Broccoli, 2015: Using Single-Forcing GCM Simulations to Reconstruct and Interpret Quaternary Climate Change. J. Climate, 28, 9746–9767, https://doi.org/10.1175/JCLI-D-15-0329.1

Lord, N. S., Crucifix, M., Lunt, D. J., Thorne, M. C., Bounceur, N., Dowsett, H., O'Brien, C. L., and Ridgwell, A.: Emulation of Long-Term Changes in Global Climate: Application to the Late Pliocene and Future, Climate of the Past, 13, 1539–1571, https://doi.org/10.5194/cp-13-1539-2017, 2017.

Lunt, D. J., Bragg, F., Chan, W.-L., Hutchinson, D. K., Ladant, J.-B., Niezgodzki, I., Steinig, S., Zhang, Z., Zhu, J., Abe-Ouchi, A., de Boer, A. M., Coxall, H. K., Donnadieu, Y., Knorr, G., Langebroek, P. M., Lohmann, G., Poulsen, C. J., Sepulchre, P., Tierney, J., Valdes, P. J., Dunkley Jones, T., Hollis, C. J., Huber, M., and Otto-Bliesner, B. L.: DeepMIP: Model intercomparison of early Eocene climatic optimum (EECO) large-scale climate features and comparison with proxy data. Submitted to Clim. Past.

Marzocchi, A., Lunt, D. J., Flecker, R., Bradshaw, C. D., Farnsworth, A., and Hilgen, F. J.: Orbital control on late Miocene climate and the North African monsoon: insight from an ensemble of sub-precessional simulations, Clim. Past, 11, 1271-1295, doi:10.5194/cp-11-1271-2015, 2015.

Rugenstein, M., Bloch-Johnson, J., Gregory, J., Andrews, T., Mauritsen, T., Li, C., et al ( 2019). Equilibrium climate sensitivity estimated by equilibrating climate models. *Geophysical Research Letters*, 46. https://doi.org/10.1029/2019GL083898

Simon, D., A. Marzocchi, R. Flecker, D.J. Lunt, F.J. Hilgen, P. Th. Meijer, Quantifying the Mediterranean freshwater budget throughout the late Miocene: New implications for sapropel formation and the Messinian Salinity Crisis, Earth and Planetary Science Letters, 472, 25-37, 2017.

Singarayer, JS & Valdes, PJ 2010, 'High-latitude climate sensitivity to ice-sheet forcing over the last 120 kyr', *Quaternary Science Reviews*, vol. 29, no. 1-2, pp. 43 - 55. https://doi.org/10.1016/j.quascirev.2009.10.011

Valdes, P. J., Armstrong, E., Badger, M. P. S., Bradshaw, C. D., Bragg, F., Crucifix, M., Davies-Barnard, T., Day, J. J., Farnsworth, A., Gordon, C., Hopcroft, P. O., Kennedy, A. T., Lord, N. S., Lunt, D. J., Marzocchi, A., Parry, L. M., Pope, V., Roberts, W. H. G., Stone, E. J., Tourte, G. J. L., and Williams, J. H. T.: The BRIDGE HadCM3 family of climate models: HadCM3@Bristol v1.0, Geosci. Model Dev., 10, 3715-3743, 2017.

---

## Author Comment (AC1) · 17 Apr 2020

*Dear editor,*

*Please find enclosed our detailed responses to the minor revisions suggested by both reviewers. Most of the figures have been reworked to include labels, following suggestions by D. Lunt. New figures have been added as well (Fig. 1; Fig. 14; Fig. 16; Fig 17; Fig. 18; Fig. 20). We have moved the performance section in the appendixes as suggested by rev#1. We provide several new references and have reworked the introduction to fit with both reviewers suggestions. We benefited from the help of J.-L. Dufresne, who is now a co-author of the paper. The detailed responses can be found below.*

*Best regards,*

*Pierre Sepulchre on behalf of co-authors.*

**Anonymous Referee #1**
**Overall Comments:**

I applaud the authors for this very impressive and detailed work. The paper covers a lot of ground and will appeal to a diverse audience including computational scientists and software engineers (technical and performance analysis), model developers (tuning decisions and model details), climate scientists (model validation against observations and similar earth system models), and finally, paleoclimate modellers and developers (deep time case study). I do have a number of suggestions, however, which are detailed below. Primarily, they are mostly minor, but important to improve the readability of this manuscript. Inconsistencies in organization and figures must be addressed before full publication.

We thank referee #1 for these comments. In the following we have replied to every comment and suggestion made, and modified the manuscript accordingly.

As the focus in main paper on scientific improvements, the authors may want to consider moving the performance-related discussion to the appendix, and climate state figures back into the main text. Another idea might be to keep the appendix material themed; something like (a) performance analysis and (b) modes of variability. Although both of these topics are very important, if the main paper covers the main model improvements and climate state analysis, it may help to tighten up the manuscript.

We agree. We have moved the performance section to the appendix and reorganized the whole section.

Lastly, although the authors note that the case study is not meant to be an evaluation of the IPSL model with the Cretaceous time period, I would like to see more discussion on how it compares to proxy data (or other published work).

Our Cretaceous simulation is only intended for illustrating the steps that were necessary to adapt IPSL-CM5A2 to the deep-time and to create deep-time boundary conditions. We refer the reader to the paper by Laugié et al. (2019) for a complete comparison of the Turonian climate simulated with IPSL-CM5A2 with reconstructions inferred from the proxy record. Laugié et al. (2019) report results from a slightly different Cretaceous simulation performed with

IPSL-CM5A2. The Laugié simulation uses the same land-sea mask and boundary conditions than we do but prescribes an estimation of the tidally-driven mixing associated with the dissipation of internal wave energy for the M2 tidal component. In addition, our Cretaceous simulation uses different values in the parameterization of the dissipation time of atmospheric waves, which were originally prescribed for numerical stability at the start of the simulation but were not corrected. These values make our Cretaceous simulation about 1 to 1.5°C warmer than the Laugié simulation, which uses appropriate values in the parameterization.

**Specific Comments:**
Introduction: Somewhere in the introduction, the acronym IPSL should be defined.
Done.

Lines 20-66: The CESM system has been run for multi-millennia for CCSM3, and now, currently with CESM1 under the TraCE project. Although this is not continuous in that the model is stopped, adjusted, and restarted for separate timeslices, it is a multi-millennia simulation nonetheless. For CCSM3:
http://www.cesm.ucar.edu/working_groups/Paleo/documentation/sims-projects/trace-21ka/ and Feng He Phd dissertation (U. Wisconsin, 2011); for CESM1, contact Dr.He.
Thank you for this suggestion. We now mention this work, and also cite the recent Kutzbach et al. paper (PNAS, 2020), that uses acceleration techniques to mimic multi-millenial simulations with CCSM3. This section has also been totally reworked following reviewer #2 suggestions.

Line 39: Define "branched". Even for climate modellers, this term can be a source of confusion.
Done. The sentence now reads: "Later, experimental designs with GCMs for deep-time paleoclimate modelling relied on simulations branched on each other (i.e. final-state of equilibrated simulations is used as initial state for new simulations with a different forcing) to make the ocean reach different equilibria in a reasonable amount of time […]"

Paragraph Line 67: For those not familiar with the IPSL model, a better overview of the original model is needed for the introductory context, i.e. a mention of all model components (are there not sea ice and river runoff in NEMO?). The Dufresne et al reference is not enough.
We now mention each component in this section. We describe later in the text the different components of both models. In particular, we have added a more thorough description of LIM2, that now reads : "*LIM2 (Fichefet and Maqueda, 1997; Timmermann et al., 2005) is a sea-ice model with a single sea-ice category. Open water is represented through ice concentration. As stated in (Uotila et al., 2017) who provide a detailed comparison between LIM2 and LIM3 in the 170 NEMO, LIM2 implements the snow-ice formation by infiltration and freezing of seawater into snow when deep enough. The effect of subgrid-scale snow and ice thickness distributions is implicitly parameterised by enhancing the conduction of heat through the ice and by melting the ice laterally to account for thin ice melting. The surface albedo depends on the state of the surface (frozen or melting), snow depth and ice thickness following (Shine and Henderson-Sellers, 1985).*"
We also added a table with the main characteristics of the model components.

From IPSL-CM5A to IPSL-CM5A2:

Section 2.1. It would be very useful to see a table summarizing this information including how each component has changed (or not changed), including resolutions.

We agree. The new version now has such a table.

Table 1: What is the significance of the bolded information? Please note in the table caption. Also, JOILOT-CURIE has not been mentioned yet. Adding the heading"supercomputer" to this column would help.

The bolded information highlights the configuration that has been selected. The table and its caption have been corrected.

Sections 2.2 and 2.3. These sections would work just as well as an appendix.

They are now part of the appendix.

Tuning

Table 2 needs units. How are the Trenberth estimates used here?

Table 2 is now corrected. Trenberth values were used as a baseline to evaluate our model results, but we mistakenly omitted them in the submitted manuscript. It is now corrected.

Lines 268 and 280: change "insure" to "ensure".

Done.

**Comparison with observations**

Line 323. There is no figure showing spatial biases, so please add "not shown".

Done.

Line 351. "every region..."

Corrected.

Line 377, Figures 7, 8, 9. It isn't clear when you are referring to Figure 7 or 8, or Figure7 or 9 in the text, thus making these sections hard to follow. Also, Line 377 refers to Fig 15 15, when I think the authors mean Fig 9. Please review text and figures for consistency. Also, please make IPSL-CMA titles in the figures consistent with one another. Figs 8/9uses "LR" when others do not.

Corrected.

Figure 10. This is more than just a Hovmueller (only panels c and d).

Corrected.

Paragraph Line 396 and Figure 11.    Consistently  refer  to  figures  here  as  in  the manuscript, i.e.  use  the  lettering  a,b,c,d.   Also,  please  use  the  dataset  name  in  the  figure

caption, rather than simply "data". For surface temperature, I assume this is HadiSST? What is used for observation for the salinity, WOA2013? I don't see this information in text either.

The information is now given. Both temperature and salinity observation fields come from WOA2013.

Line 426- 429 and Figure 15a. The figure caption needs to be improved. It is clear that 15b, the colors are the differences from the two models, but not so with 15a. If the purpose is to highlight the improvement in AMOC for 15a, then a figure showing the timeseries of AMOC, maximum below 500, between the two models, could be an alternative figure.

We have corrected the figure and caption that were actually wrong. We now present the AMOC anomaly for 100 years of preindustrial run for both models (new figure below). We thank the reviewer for suggesting to add the time series of AMOC maximum in the two model versions. Nevertheless, there is no point in strictly comparing the time evolution of these two time series as there is no reason for internal variability in the two simulations to be phased. We thus prefer to stick with the comparison between spectra as shown on Fig. 17.

[Figure]

Section 4.2.4 and Figure 17.

Consistency between this figure and others in this manuscript regarding lettering or positional references, and labeling of models is needed. Also, the caption needs to be improved to include years and model names.

Corrected.

For the text, some discussion with observations is needed for context.

Done. We now refer to several publications to discuss the 20-yr variability.

Figure A5. What is the difference between the gray and black lines in the power spectra ? Also, labelling of appendix figures needs to be consistent with the rest of the manuscript.
The grey line shows values of the HadISST dataset for comparison.

Figure A6. Consistency in labelling.
Corrected.

**Deep time case study**

Line 503. "..ran a 3000-year long..."Section 5.1 and Figure 18.  I am not sure what I am looking at in Figure 18a.  Are the white areas supposed to be the grey-shading for the refinement? (This could be a document format issue so, perhaps noting the refinement areas with a black box rather than colored shading would be better). Also, for Figure 18b, my recommendation would be to make this with the cretaceous continental outlines, not modern. I think the discussion of pole and refinement placement would be much clearer to readers who have never modified grids for deep time if the paleo outlines were used.

We have added the Cretaceous continental outlines as recommended. The grey shading on fig 18a (now fig. 17a) depicts the fact that continental points have been used to refine the resolution over the different seas mentioned in the caption text.

[Figure]

**Section 5.2.**  What did you use for other important forcings, such as solar input and methane? Should we assume that all else is based on pre-industrial forcings?  Does this model prescribe or predict aerosol emissions? Dust, sea salt, carbon from fire, and other biogenic aerosols such as secondary organic aerosols (isoprene, etc), may also be extremely important climate forcings, even for deep time.

Thank you for pointing out these omissions. The solar constant was set at Cenomanian-Turonian value (1353.36 W.m$^{-2}$). Methane and prescribed aerosol emissions were set at preindustrial values, as this version of IPSL-CM5A2 does not predict aerosol emissions. We totally agree that aerosols emissions could be extremely important for Cretaceous climate, but computing them (if possible at all) was out of the scope of this paper.

Line 580. I suspect the "?" was meant to stand in for something? Please fix.
Fixed, it was a missing reference (to Wilson et al. Palaeo-3, 2012) in our latex file.

Line 582 and Figure 19.  Should this be Fig 19a?  Given that 19b is discussed first, a and b should be swapped in Figure 19.
Corrected. Fig 19 (now fig. 18) has been modified.

Line 592. Have the authors worked on applying the dynamic vegetation to the Cretaceous? Dynamic vegetation capabilities are highly valued in the deep time modelling community.

As in IPSL-CM5A, dynamic vegetation was not activated in these simulations, but we are currently working on implementing this feature in the near future.

Line 607 and Figure 20. Define T2M. Is it 2-meter air temperature, same as "t2m"? Please make consistent throughout the manuscript.

Yes. Corrected.

Also, I would not consider 20e a"trend" plot as the figure caption states. Although trend can certainly be implied, but the variable being plotted is a whole value as it varies through time, the unit being degC(or Joule).

We have replaced trend by time-series.

Figure 21. Assuming this is annual mean. How many years (the final 100?) went into this plot?

The mean Cretaceous climate state illustrated on Fig. 21 is indeed the annual mean, and you are correct in inferring that the final 100 years were averaged to make this plot. The caption has been modified accordingly.

Line 635. Fill in Supplemental Figure XX. Is this referring to figures A8 and A9?

Corrected.

Figure A8 and A9. Please label consistently. Briefly note/explain why you chose JMA and JAS and northern/southern hemisphere winter.

We use JFM and JAS as Northern/Southern Hemisphere winter because the maximum mixed-layer depth in (deep) convection areas occurs in March and September for the Northern/Southern Hemispheres, respectively.

**Conclusions**

Line 645. "This article aims..."I would recommend changing the title of this section to "Conclusions and Discussion"because of 1) the inclusion of new appendix material and 2) the relevance of the new material to the overall discussion on challenges for (deep time) paleoclimate modelling.

Unfortunately the title "Conclusions" is imposed in the GMD template (copernicus.cls in the Latex template).

**Appendices**

An appendix section describing the many acronyms in this paper would be helpful. Appendix organization needs to be fixed.

Fixed.

There are two A1's. Fixed.

Also, only the first A1 includes text to introduce the appendix figures. A few sentences introducing each appendix section would greatly improve the readability of this appendices.
Corrected.

**Review #2 by D. Lunt.**

I was looking forward to reviewing this paper because (a) it is exactly the sort of paper that we hoped the existence of GMD would enable and encourage when journal first started, (b) I have been involved in a similar paper describing our own model (Valdes et al, 2017), and (c) it has a section on configuring the model for deep-time simulations, which is of great interest scientifically to me. I think that this paper will be very useful, providing a definitive benchmark for this model. I hope that the authors start a Special Issue on this model, and submit bug-fixes/developments as they occur, in follow-up papers. I have some questions/comments (none of which are major), some suggestions for additional work (which I would view as nice but optional), and some minor corrections.
We thank Dan Lunt for these comments and the suggestion of starting a special issue on the model. We will consider this option very seriously. In the following we answer the different comments and describe how we addressed them in the manuscript when necessary.

**Questions/Comments:**
•Introduction: as well as paleoclimate, there will be interest from the forcings/feedbacks community associated with longrun MIP (e.g. Rugenstein et al, 2019). Would be good to mention this. In general, I think you underplay the interest of this model to the non-paleo community.
Agreed, we have modified the introduction accordingly, and included this reference.
•Abstract needs to include the resolution of the model (horizontal and vertical, atmosphere and ocean).
The resolution has been added to the abstract.

•Line 20: Fully complex(e.g. CMIP6) models don't only carry out snapshots, often transient runs (e.g. RCPs!). More generally, remember the audience for this paper is not just paleo scientists-you should assume an audience that is not familiar with paleoclimates, e.g. avoid terms such as "Quaternary", "last interglacial" without explaining when they were.
Agreed, we have defined pre-Quaternary in the introduction.

•Line 64: I would remove the statistical approach from the transient section, and instead add another two types of approach here, which are:

(a)"multiple snapshots". Within this you could include e.g. Singarayer and Valdes(2010), and Marzocchi et al(2015).

(b)statistical approaches based on multiple simulations, e.g. Lord et al (which you already cite), and Simon et al(2017), and Erb et al (2015).

We agree and have modified the text accordingly. We also include more references suggested by reviewer #1, so the entire section has been reorganised. Now it reads :

[revised manuscript text omitted]

•Line 96: It is not always clear from the description which components are identical to IPSL-CM5A, and which are "updated versions of IPSL-CM5A components".
We have now included a table specifying the differences between both versions.

•Line 116 indicates that "The differences mostly include bug corrections, various improvements in the physics/dynamics interface, and energy conservation. Changes also concern the Emanuel convection scheme, with bug corrections and changes in the upper bound of the deep convection loop.".  it would be good to maybe say a little bit more about these, and explain why these changes were made.
The purpose of this sentence was to say that the changes between LMDZ5A and LMDZ5A2 were essentially computational and had almost no effect on the model results. The sentence was simplified and modified as follows:
*"The differences mainly concern bug fixes and various improvements in the physical/dynamic interface in preparation for the next generation of models. They have very limited effect on model results and translate into a slight improvement in energy conservation."*

•Figure 1 looks pretty, but would include more information if a standard cylindrical projection were used?  Or maybe if views towards the Pacific were also added.
Yes. We have decided to keep this projection and show both hemispheres for LMDZ and NEMO.

[Figure]

•Section 2.1: I like the way that the actual tagged releases of each component are given for the new model and for the old model. This is very good practice. Is there also a version number for the overall model, for both new and old versions? And is there a unique update number for NEMO and OASIS as well (the later section on Code Availability indicates that there is)?
We now present a table with specific details on version/revision numbers.

•Line 229: Not sure what "adjusted" means in "provided an adjusted global surface air temperature of 11.3°C"
We rephrased to "A first untuned 1,000-year spin-up of IPSL-CM5A2 forced by CMIP5 pre-industrial boundary conditions provided an adjusted global surface air temperature of 11.3°C."
•Line 230: "Such a cold bias had been depicted in the previous IPSL-CM5A (Dufresne et al., 2013)". What was the magnitude of that bias ?
Dufresne et al. (2013) discuss a bias of 1K in the historical run compared to observations.

•Line 231: "We targeted to increase global-mean surface temperature by 2.2°C to reach 13.5°C in pre-industrial conditions with IPSL-CM5A2, expecting this value to translate into 15.5°C in simulations with present-day conditions.". This needs a reference to some observational dataset, either to validate the 13.5 preindustrial choice, and/or the 15.5 present-day choice. Also mention the uncertainty in these observations.
We agree. We rephrased and it now reads :
A first untuned 1,000-year spin-up of IPSL-CM5A2 forced by CMIP5 pre-industrial boundary conditions provided an equilibrated global surface air temperature of 11.3°C. Such a cold bias had been depicted in IPSL-CM5A (Dufresne et al., 2013). We targeted to increase global-mean surface temperature by 2.2°C to reach 13.5°C in pre-industrial conditions with IPSL-CM5A2, expecting this value to translate into 14.5°C for a present-day simulation, a value consistent with observations (Dee et al., 2011).

•Line 236: "Previous results obtained with LMDZ and IPSL-CM5A show that changing TOA balance by 1 W.m$^{-2}$ results in a change in temperature of 1 K." I guess you mean "initial TOA inbalance"?
We actually mean TOA balance. We follow the protocol described in Hourdin et al, 2017 : "A common practice to fulfill this constraint is to adjust the top-of-atmosphere or surface energy balance in atmosphere-only simulations exposed to observed sea surface temperatures (component tuning) and check if the temperature obtained in coupled models is realistic. This energy balance tuning is crucial since a change by 1 W m$^{-2}$ of the global energy balance typically produces a change of about 0.5–1.5 K in the global-mean surface temperature in coupled simulations depending on the sensitivity of the given model."

Is this the TOA in a PI run, or historical, or....?
The TOA is that of the preindustrial run.

•Line 237: "It is also the typical value of the sensitivity of global temperature to greenhouse gas concentration" is rather opaque....do you mean that the overall net feedback parameter, lambda, in this model is ~1.0 W/m2 / K ?

Agreed, we removed this sentence.

•Line 249: "when two of the bugs mentioned above"...there are several bugs mentioned above...which ones are you referring to?

The paragraph has been corrected to specify the two bugs.

•Line 257: "respectively 0.029 W.m-2 and 0.023 W.m−2".  Is the difference between these two primarily due to interannual variability, or gradual accumulation of energy in the ocean, or bugs in energy conservations, or something else?

This difference is most likely linked to the fact that the model is not fully energy-conservative.

•Line 258: What are the causes of the non-conservation of energy?

The non conservation of energy in the model has several causes. Conversion terms between potential energy and kinetic energy and dissipation of kinetic energy to heat are not fully considered (both in the atmosphere and in the ocean). The energy fluxes into the ocean associated with rain and snow, riverflow and icebergs are also approximated.

In very long simulations, small non-conservation of energy is not a problem, because the energy system is "open" to the outside –i.e. a small non-conservation may have very similar results to slightly increasing the solar constant –the model just finds a new equilibrium.  However, the hydrological system is "closed", and so non-conservation can result in substantial drifts in e.g. global mean salinity over very longtime periods.  How well does the model conserve water, and are any correction factors included to maintain global mean water/salinity content? OK, I see this is addressed later (lines 272-280).  Did you consider a salinity correction term?  See e.g. the HadCM3 runs in DeepMIP –Lunt et al (submitted to Climate of the Past).

We have not yet considered a salinity conservation term because the non conservation is linked to the LIM2 sea ice model.  IPSL-CM5A2 will likely be upgraded to include the fully water-conservative LIM3 sea ice model in the future but, in the short term, we expect IPSL-CM5A2 to be mostly used for simulating warm climates, thereby reducing any potential salinity drift.

•Line 268: "Such a difference is also small enough compared to the regional biases in sea surface temperature." How do you define "small enough"?! Would 0.5 degrees by small enough?  What about 0.8, or 1.0?! Better to compare with some other models, and just say, e.g. "this is closer to observations than X% of CMIP5 models", or(and)link it to the uncertainty in the observations –0.3 degrees is probably within error of the observations?

We agree this sentence is unclear, and can be misleading. We have simply removed it.

•Supp Info could be used for some more plots, e.g. the old model version of Figure 4(d,e),the old model version of Figure 5(c).6(a,d) etc etc.  I expect you have already made these plots!
Yes, we agree it would be valuable to have these plots in the appendixes, but the paper is already very long. Instead we refer to the online atlases comparing both versions of the model.

•Label all figures, not only in the caption but above the figure panels as well.  Some plots are model minus observations, some model minus model, some just model. Clearere labelling would help (I know I could get this info from the caption but this is much slower than labelling the panels!)
Most of the figures have been modified to include labels.

•Fugure 15a –presumably this is the new model (not stated in the caption).  Would be good to see this also for the old model.
This figure has been modified and now presents the anomaly between IPSL-CM5A2 and IPSL-CM5A both for the AMOC and the barotropic streamfunction.

•Line 595: would be good to show evolution of TOA flux, and show a Gregory plot.
 We now include both plots in the appendixes section.

[Figure]

[Figure]

•Section 5.4.  This section implies that the model worked "first-time", "out-of-the-box" for the new Cretaceous paleogeography.  I would be surprised (and impressed!) if this were the case.  If not, then it would be good to explain what needed to be done to allow the model to run, e.g. smoothing topography/bathymetry etc.

This section is intended to review the issues that were hampering the use of earlier versions of the IPSL model for deep-time simulations and to present the major steps required to set up a deep-time paleoclimate simulation in IPSL-CM5A2. Most of the locks preventing the model to run indeed concerned the grid structure (e.g. locations of the poles) and the boundary conditions.

As most other models, some tweaks are necessary for IPSL-CM5A2 to run with very different configurations. For instance, numerical instabilities require that the timestep of NEMO is slightly increased. In the atmosphere, warmer climates due to increased atmospheric CO2 levels regularly make the convection scheme crash. We usually circumvent these instabilities by adding white noise to surface pressure values in the restart files or changing atmospheric wave dissipation time in the model. By increasingly using IPSL-CM5A2 in a deep-time configuration, we have observed that this latest method greatly reduces the number of crashes but that the model is not tuned for running with these dissipation values. As a result, the Cretaceous simulation that we report here (being one of the first ever made with the deep-time version of IPSL-CM5A2) is 1 to 1.5°C warmer than the similar Cretaceous simulation reported by Laugié et al. (2019), which has been performed later and runs with appropriate dissipation values (see also response to reviewer #1). However, polar smoothing applied to topography/bathymetry was not needed for the model to run.

**Suggestions**

•Line 282: Presumably there is quite a "kick" to the model going from your PREIND forcings to the 1850 forcings; presumably it might have been a good idea to hold the model steady at 1850 CMIP5 forcings for a few decades before starting the 1850-2005 simulation?

Actually,the model was run for thousands of years in preindustrial conditions before. Then the restarts of PREIND simulation have been used to initialize the historical simulation.

•HadCM3 runs at about 100 SYPD these days (see Valdes et al, 2017).  it might be nice to compare some metrics with them, in a consistent style e.g. Geckler plots (see Figure 2 in Valdes et al, 2017).

We agree and we think it could indeed be the scope of a dedicated paper, comparing the performance of fast low resolution climate models. For information we still provide a parallel coordinate graph comparing IPSL-CM5A2 metrics to other CMIP5 models, and highlighting HadCM3 as well. The variables are sorted to display the results by increasing order of performance for IPSL-CM5A (black line). IPSL-CM5A2 (blue line) displays improvements in temperature fields, whereas precipitation is slightly worsened. HadCM3 (CMIP5 version) is displayed in red, showing better performances in wind fields and precipitation, and worse in temperature. We did not have the values for HadCM3BL, which would be more relevant to compare to IPSL-CM5A2.

[Figure]

•In addition to the Cretaceous simulation with the new model grid, it would be very informative to re-do a preindustrial control with this new grid, to see how much of a role the modern-specific tweaks play

We agree and plan to do this simulation in the near future.

**Minor corrections**

Line 8: "and" missing.

corrected.

Line 27: "which" should be "whose"?

rephrased.

Line 150: spell out "phytoplankton"and "zooplankton".

corrected.

Figure 2 caption: "The surface of the disks" should be "The surface area of the disks".

corrected.

Figure 4: label panels (d) and (e)as CRF and temp anomalies (just so we don't have to read the caption!).

Figure 19b. Black continental outline is missing for some coastlines.

corrected.

Line 580.  Reference missing.
Now included.

Figure 21: why suddenly a new map projection?
We provide new versions of this figure (now figure 20), consistent with the projections used earlier in the manuscript.
I was surprised that Jean-Louis Dufresne was not a co-author, given his contributions to the original model?
J-L Dufresne has been of great help contributing to the revised version of the manuscript, and is now a co-author.

All the following suggested references are now cited in the text.
Erb, M.P., C.S. Jackson, and A.J. Broccoli, 2015: Using Single-Forcing GCM Simulations to Reconstruct and Interpret Quaternary Climate Change. J. Climate, 28, 9746–9767, https://doi.org/10.1175/JCLI-D-15-0329.1
Lord, N. S., Crucifix, M., Lunt, D. J., Thorne, M. C., Bounceur, N., Dowsett, H., O'Brien, C. L., and Ridgwell, A.: Emulationof Long-Term Changes in Global Climate: Application to the Late Pliocene and Future, Climate of the Past, 13, 1539–1571,https://doi.org/10.5194/cp-13-1539-2017, 2017.

Lunt, D. J., Bragg, F., Chan, W.-L., Hutchinson, D. K., Ladant, J.-B., Niezgodzki, I., Steinig, S., Zhang, Z., Zhu, J., Abe-Ouchi, A., de Boer, A. M., Coxall, H. K., Donnadieu, Y., Knorr, G., Langebroek, P. M., Lohmann, G., Poulsen, C. J., Sepulchre, P., Tierney, J., Valdes, P. J., Dunkley Jones, T., Hollis, C. J., Huber, M., and Otto-Bliesner, B. L.: DeepMIP: Model intercomparison of early Eocene climatic optimum (EECO) large-scale climate features and comparison with proxy data. Submitted to Clim. Past.

Marzocchi, A., Lunt, D. J., Flecker, R., Bradshaw, C. D., Farnsworth, A., and Hilgen, F. J.: Orbital control on late Miocene climate and the North African monsoon: insight from an ensemble of sub-precessional simulations, Clim. Past, 11, 1271-1295, doi:10.5194/cp-11-1271-2015, 2015.

Rugenstein, M.,Bloch-Johnson, J.,Gregory, J.,Andrews, T.,Mauritsen, T.,Li, C., et al (2019).Equilibrium climate sensitivity estimated by equilibrating climate models.Geophysical Research Letters,46.https://doi.org/10.1029/2019GL083898

Simon, D., A. Marzocchi, R. Flecker, D.J. Lunt, F.J. Hilgen, P. Th. Meijer, Quantifying the Mediterranean freshwater budget throughout the late Miocene: New implications for sapropel formation and the Messinian Salinity Crisis, Earth and Planetary Science Letters, 472, 25-37, 2017.

Singarayer, JS& Valdes, PJ2010, 'High-latitude climate sensitivity to ice-sheet forcing overthe last 120 kyr',Quaternary Science Reviews, vol. 29, no. 1-2, pp. 43 -55.https://doi.org/10.1016/j.quascirev.2009.10.011

Valdes, P. J., Armstrong, E., Badger, M. P. S., Bradshaw, C. D., Bragg, F., Crucifix, M., Davies-Barnard, T., Day, J. J., Farnsworth, A., Gordon, C., Hopcroft, P. O., Kennedy, A. T., Lord, N. S., Lunt, D. J., Marzocchi, A., Parry, L. M., Pope, V.,Roberts, W. H. G., Stone, E. J., Tourte, G. J. L., and Williams, J. H. T.: The BRIDGE HadCM3 family of climate models: HadCM3@Bristol v1.0, Geosci. Model Dev., 10, 3715-3743, 2017.